# Post-transcriptional repression of circadian component CLOCK regulates cancer-stemness in murine breast cancer cells

**Takashi Ogino[1†], Naoya Matsunaga[1,2†], Takahiro Tanaka[1], Tomohito Tanihara[1], Hideki Terajima[3], Hikari Yoshitane[3], Yoshitaka Fukada[3], Akito Tsuruta[1], Satoru Koyanagi[1,2], Shigehiro Ohdo[1]\***

[1]Department of Pharmaceutics, Faculty of Pharmaceutical Sciences, Kyushu University, Fukuoka, Japan; [2]Department of Glocal Healthcare Science, Faculty of Pharmaceutical Sciences, Kyushu University, Fukuoka, Japan; [3]Department of Biological Sciences, School of Science, The University of Tokyo, Tokyo, Japan

**Abstract** Disruption of the circadian clock machinery in cancer cells is implicated in tumor malignancy. Studies on cancer therapy reveal the presence of heterogeneous cells, including breast cancer stem-like cells (BCSCs), in breast tumors. BCSCs are often characterized by high aldehyde dehydrogenase (ALDH) activity, associated with the malignancy of cancers. In this study, we demonstrated the negative regulation of ALDH activity by the major circadian component CLOCK in murine breast cancer 4T1 cells. The expression of CLOCK was repressed in high-ALDH-activity 4T1, and enhancement of CLOCK expression abrogated their stemness properties, such as tumorigenicity and invasive potential. Furthermore, reduced expression of CLOCK in high-ALDH-activity 4T1 was post-transcriptionally regulated by microRNA: miR-182. Knockout of miR-182 restored the expression of CLOCK, resulted in preventing tumor growth. Our findings suggest that increased expression of CLOCK in BCSCs by targeting post-transcriptional regulation overcame stemness-related malignancy and may be a novel strategy for breast cancer treatments.

**\*For correspondence:**
ohdo@phar.kyushu-u.ac.jp

[†]These authors contributed equally to this work

**Competing interests:** The authors declare that no competing interests exist.

## Introduction

Breast cancer is the most common cancer in women globally. There are many diagnostic and therapeutic strategies developed for the treatment of breast cancer; however, recurrence and metastasis remain causes of poor prognosis in breast cancer patients. These problems are thought to be associated with tumor heterogeneity, as tumors are composed of heterogeneous cells with structural and biochemical differences (*Anderson et al., 2011*; *Burrell et al., 2013*; *Ding et al., 2012*). Some specific tumor cell populations present as cancer stem-like cells (CSCs) have a high malignant potential and resistance to chemotherapy, due to their high differentiation and self-renewal capacities (*Bonnet and Dick, 1997*; *Meacham and Morrison, 2013*; *Patrawala et al., 2006*; *Visvader and Lindeman, 2008*). Therefore, targeting CSCs may improve breast cancer prognosis, but remains challenging due to the malignant properties of CSCs.

Breast cancer stem-like cells (BCSCs) were first reported in solid tumors and specific BCSC markers such as cell surface antigens CD44/CD24 have been identified to isolate stem-like cell populations (*Al-Hajj et al., 2003*). Aggressive breast tumors like triple-negative breast cancers (TNBCs) are composed of large populations of cells with high aldehyde dehydrogenase (ALDH) activity (*Charafe-Jauffret et al., 2009*; *Ginestier et al., 2007*). Evidence suggests that breast cancer cells exhibiting high ALDH activity are associated with poor prognosis and late-stage tumors (*Charafe-*

*Jauffret et al., 2010*; *Marcato et al., 2011*); ALDH is now also considered as a CSC marker for other types of cancers (*Jiang et al., 2009*; *Li et al., 2010*; *Pearce et al., 2005*).

Our recent study demonstrated that the number of cells with high ALDH activity exhibits circadian oscillation in 4T1 breast cancer tumors implanted in mice (*Matsunaga et al., 2018*). A time-dependent variation in the number of cells with high ALDH activity cells affected the anti-tumor effects of chemotherapeutic drugs. Molecular circadian oscillators consisting clock genes induce time-dependent changes in the chemosensitivity of cancer cells by controlling the expression of cell cycle regulators and apoptotic factors (*Horiguchi et al., 2013*). However, we also found that oscillation in the expression of clock genes was suppressed in ALDH-positive CSCs but not in the surrounding cells in the tumor microenvironment (*Matsunaga et al., 2018*). WNT10a rhythmically secreted from the microenvironment stimulates ALDH-positive CSCs in a time-dependent manner, and this activation of the WNT/β-catenin signaling in CSCs results in a rhythmic expression of ALDH.

Studies have reported the association of clock gene dysfunction with cancer malignancy (*Katamune et al., 2019*; *Katamune et al., 2016*; *Masri et al., 2015*). Notably, oncogenic transformation of *Period2* circadian gene-defective cells results in the development of chemoresistance (*Katamune et al., 2019*), suggesting the relevance of circadian clock disruption in the maintenance of CSC properties. However, the role of the circadian clock system in the regulation of CSC biology needs to be explored further.

In this study, we elucidated the role of each circadian clock gene in the maintenance of CSC properties of mouse 4T1 breast cancer cells. Among the clock genes, expression levels of *Clock* mRNA and its protein were downregulated in ALDH-positive 4T1 cells, and transduction of CLOCK-expressing lentivirus into the ALDH-positive 4T1 cells attenuated their malignant potential. Further, we investigated the underlying mechanism of CLOCK downregulation in ALDH-positive 4T1 cells. Our findings present a possible strategy to overcome the malignancy of BCSCs by targeting miRNA-mediated post-transcriptional regulation of circadian component CLOCK.

## Results

### Expression of CLOCK is suppressed in ALDH-positive 4T1 cells

CSCs are often characterized by high ALDH activity associated with malignancy and is used for identification and isolation of CSCs (*Ginestier et al., 2007*; *Ma and Allan, 2011*; *Nagare et al., 2016*). As reported previously (*Kim et al., 2013*), ALDEFULOR assay revealed the presence of a high ALDH activity (ALDH-positive) cell population among the 4T1 breast cancer cells. To confirm the stem-like properties of the isolated ALDH-positive 4T1 cells, the separated cells were plated under spheroid-forming conditions. The ALDH-positive 4T1 cells were capable of forming tumor spheroids ($p < 0.01$, *Figure 1A*), whereas such spheroid formations were not observed by ALDH-negative 4T1 cells, suggesting characterization of CSC-like properties by high ALDH activity in 4T1 cells.

Next, we investigated whether the clock gene product affected the ALDH activity of 4T1 cells. To achieve this, ALDH-positive 4T1 cells were collected by ALDEFULOR assay, and then transfected with plasmid vectors expressing products of the circadian genes, BMAL1, CLOCK, CRY1, PER1, or PER2 and these expression vectors sufficiently increased the levels of circadian gene transcripts (*Figure 1—figure supplement 1*). Among the gene products, CLOCK and PER2 significantly reduced the ratio of the ALDH-positive cell population to the ALDH-negative cell population ($p < 0.01$ respectively, *Figure 1B*); but the effect was more potent in CLOCK-transfected 4T1 cells. As we reported previously, the expression levels and circadian oscillation of *Clock* and *Per2* mRNA were disrupted in 4T1 tumor bearing mice (*Figure 1C*). In particular, the expression levels of *Clock* mRNA were lower in ALDH-positive cells than ALDH-negative cells throughout the day. Therefore, we further focused on the role of the *Clock* gene in ALDH-positive 4T1 cells.

CLOCK acts as a transcriptional factor by binding to E-box (CACGTG) elements to increase the expression of target genes (*Gekakis et al., 1998*; *Hogenesch et al., 1998*; *Ueda et al., 2005*). The levels of *Clock* mRNA and protein were significantly lower in ALDH-positive 4T1 cells than in ALDH-negative 4T1 cells in vitro ($p < 0.05$, *Figure 1D*); also E-box driven luciferase activity was significantly reduced ($p < 0.01$, *Figure 1E*). Furthermore, increased CLOCK expression by lentiviral transduction enhanced oscillation of E-box driven luciferase bioluminescence in ALDH-positive 4T1 cells ($p < 0.05$,

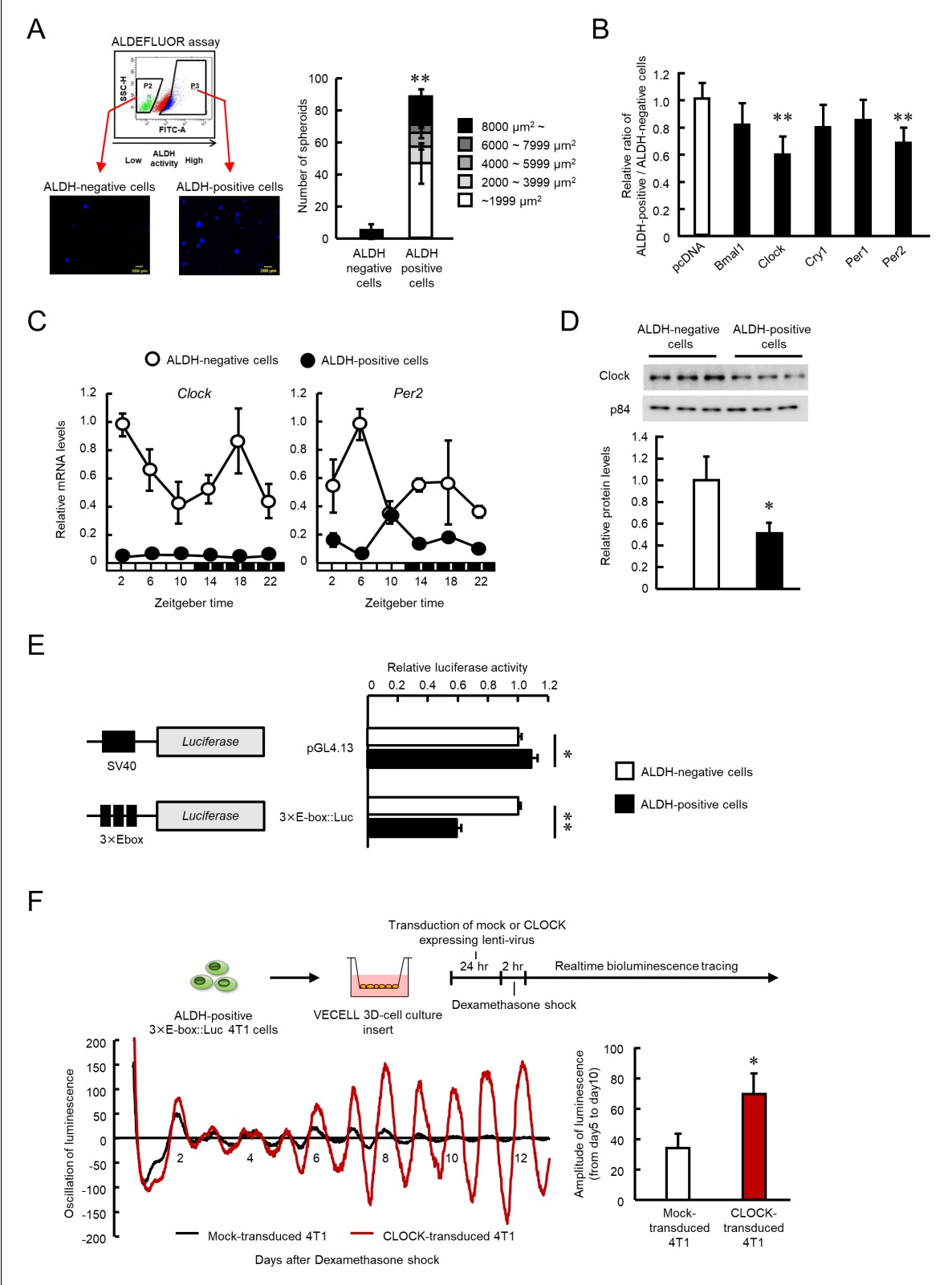

**Figure 1.** The role of CLOCK in the regulation of ALDH activity in 4T1 mouse breast cancer cells. (**A**) Representative photograph of Hoechst-stained tumor spheroids of ALDH-negative or -positive 4T1 cells in agarose. Values are the mean with SD (n = 3). **p<0.01; significant difference from ALDH-negative cells ($t_6$ = 12.598, Student's t-test). (**B**) Influence of each clock gene expression vector on the ratio of ALDH positive- to negative-cell populations. ALDH-positive 4T1 cells were cultured on 3D scaffold chambers after being transiently transfected with each expression vector using

*Figure 1 continued on next page*

*Figure 1 continued*

electroporation. ALDH activity was evaluated 5 days after transfection. Values are the mean with SD (n = 6). The mean value of the pcDNA group is set at 1.0. **p<0.01; significant difference from the pcDNA group ($F_{5,30}$ = 6.807, p<0.001, ANOVA, Dunnett's post hoc test). (C) The temporal mRNA expression profiles of *Clock* and *Per2* in ALDH-positive and ALDH-negative cells isolated from 4T1 tumor-bearing mice kept under the light/dark cycle (zeightgber time 0. lights on ; zeitgeber time 12. lights off). Values are the mean with SD (n = 3). Data were normalized by *β-Actin* mRNA levels. (D) Difference in the expression levels of CLOCK protein between ALDH-negative and -positive 4T1 cells. The value of ALDH-negative cells is set at 1.0. Values are the mean with SD (n = 3). *p<0.05; significant difference between two groups ($t_4$ = 3.915, Student's t-test). (E) Difference in the promoter activities of E-box-driven luciferase reporter in ALDH-negative and -positive 4T1 cells. pGL4.13 or 3 × E-box::Luc reporter vectors were transfected into 4T1 cells, and luciferase assay was performed after cell sorting. Values are the mean with SD (n = 3). The value of ALDH-negative cells is set at 1.0. **p<0.01, *p<0.05; significant difference between two groups ($t_4$ = 2.941 for pGL4.13, t4 = 43.287 for 3 × E-box::Luc, Student's t-test). (F) The influence of lenti-viral CLOCK transduction on the circadian oscillation of E-box-driven luciferase bioluminescence. Top panel shows the scheme of experimental procedure. Bottom panels show real-time bioluminescence tracing of luciferase activity after dexamethasone synchronization (left) and mean of amplitude of bioluminescence oscillation from day 5 to day 10 after synchronization (n = 3) (right). *p<0.05; significant difference between two groups ($t_4$ = 3.691, Student's t-test).

The online version of this article includes the following source data and figure supplement(s) for figure 1:

**Source data 1.** This spreadsheet contains the source for *Figure 1*.

**Figure supplement 1.** The mRNA expression of clock genes in 4T1 cells transfected with vectors expressing *Bmal1, Clock, Cry1, Per1,* or *Per2*.

**Figure supplement 1—source data 1.** This spreadsheet contains the source for *Figure 1—figure supplement 1*.

*Figure 1F*). These findings suggest a negative correlation between CLOCK function and stemness-related properties in 4T1 cells.

## CLOCK-mediated suppression of 4T1 cell malignancy

We investigated if an increase in CLOCK levels alters the malignant properties of 4T1 cells, by infecting 4T1 cells with CLOCK expressing lentivirus. Transduction of the *Clock* gene significantly increased its protein levels in 4T1 cells (p<0.01, *Figure 2A*). Flow-cytometric analysis revealed a reduction in the population of ALDH-positive 4T1 cells by transduction of the *Clock* gene (p<0.01, *Figure 2B*). Nine isoforms of ALDH have been confirmed to be active in ALDEFLUOR assay (*Zhou et al., 2019*); however, only the expressions of *Aldh3a1, 3a2, 3b1,* and *5a1* were detected in 4T1 cells. Among them, expression levels of *Aldh3a1* were considerably decreased in CLOCK-expressing 4T1 cells (*Figure 2C*), suggesting repression of ALDH activity by CLOCK in 4T1 cells through the downregulation of the isoform *Aldh3a1*. Database analysis of the mouse *Aldh3a1* gene upstream region led to identification of CCAAT/enhancer binding protein α (C/EBPα) as a mediator of CLOCK-controlled expression of ALDH (*Figure 2—figure supplement 1A*). There was increase of C/EBPα expression in enhanced CLOCK-expressing 4T1 cells (*Figure 2—figure supplement 1B*). Transfection of 4T1 cells with C/EBPα-expressing vectors decreased gene promoter activity and protein levels of Aldh3a1 (*Figure 2—figure supplement 1D and E*), and knockdown of C/EBPα increased protein levels of Aldh3a1 (*Figure 2—figure supplement 1F*). These results suggest that C/EBPα has important roles on CLOCK-mediated ALDH suppression, through transcriptional repression of *Aldh3a1* gene in 4T1 cells. A significant decrease in the mRNA levels of stemness related factors *Klf4, Nanog,* and *Myc* was also noted in the ALDH-positive population of enhanced CLOCK-expressing 4T1 cells (p<0.01, *Figure 2D*). Additionally, growth ability and spheroid formation of enhanced CLOCK-expressing 4T1 cells were lower than those of mock lentivirus-transduced cells (p<0.01, *Figure 2E and F*).

Malignancy of CSCs is characterized by their invasive and metastatic capacities (*Polyak and Weinberg, 2009*; *Singh and Settleman, 2010*; *Weidenfeld and Barkan, 2018*), hence we conducted two types of invasion assays in vitro. TGFβ1-induced invasion of cancer cells into collagen type 1 was assessed using a 3D cell culture chip, and the invasive potential of enhanced CLOCK-expressing 4T1 cells was lower than that of Mock-transduced 4T1 cells (p<0.01, *Figure 3A*). The low invasive potential of enhanced CLOCK-expressing 4T1 cells was also observed in the spheroid invasion assay (*Figure 3B*). Mock-transduced 4T1 cells formed spheroids that invaded into the hydrogels, whereas no invasion by CLOCK-expressing 4T1 cells-formed spheroids was detected. One of the main processes of invasion is epithelial-mesenchymal transition (EMT), represented by decrease in epithelial adherent molecules and increase in mesenchymal adherent molecules (*Nieto et al., 2016*). There was an increase in mRNA and protein levels of epithelial molecules, E-cadherin, and Claudin1 in

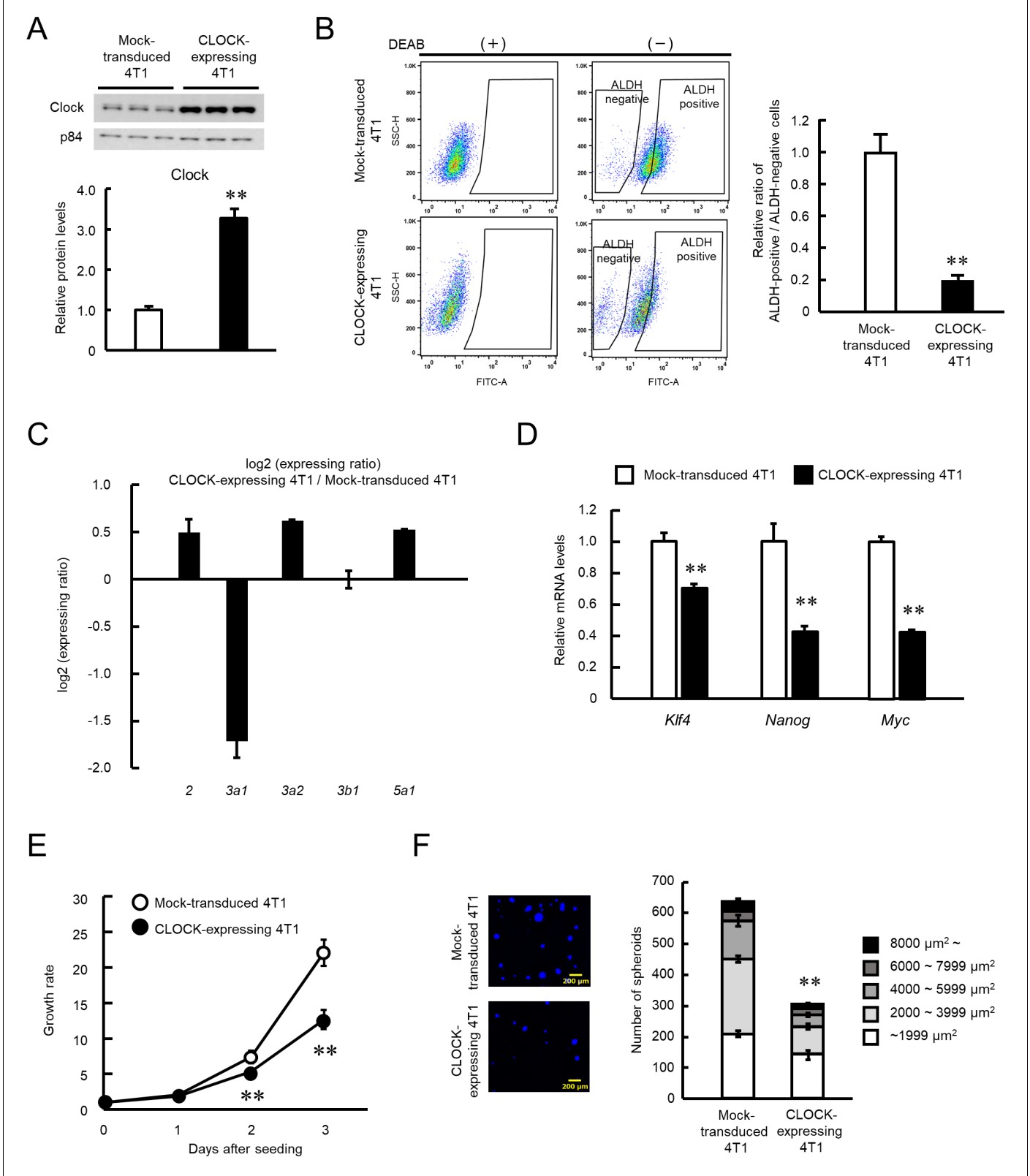

**Figure 2.** Suppression of stemness properties of 4T1 cells by CLOCK. (**A**) CLOCK protein levels in mock-transduced and *Clock*-expressing lentivirus-infected 4T1 cells. Values show mean with SD (n = 3). **p<0.01; significant difference from mock-transduced 4T1 cells ($t_4$ = 15.736, Student's t-test). (**B**) The flow cytometric analysis of ALDEFLUOR assay of CLOCK-expressing 4T1 cells. The population of ALDH-positive cells was defined based on each

*Figure 2 continued on next page*

*Figure 2 continued*

DEAB group, whose means of fluorescence intensity were set almost the same. The right panel shows the difference in the ratio of ALDH positive- to negative-cell populations between mock-transduced and CLOCK-expressing 4T1 cells. Values show the mean with SD (n = 6). **p<0.01; significant difference from mock-transduced 4T1 cells ($t_{10}$ = 22.455, Student's t-test). (C) Difference in the mRNA levels of ALDH isoforms between mock-transduced and CLOCK-expressing 4T1 cells. Values show the mean with SD (n = 3). (D) The mRNA levels of stemness-related genes in ALDH-positive mock-transduced and CLOCK-expressing 4T1 cells. Data were normalized by the *18s rRNA* levels. Values show the mean with SD (n = 3). The values of the mock-transduced group are set at 1.0. **p<0.01: significant difference from mock-transduced 4T1 cells ($t_4$ = 9.261 for *Klf4*; $t_4$ = 8.001 for *Nanog*; $t_4$ = 32.576 for *Myc*; Student's t-test). (E) Difference in growth ability between mock-transduced and CLOCK-expressing 4T1 cells. Values show the mean with SD (n = 5–6). Cell viability of seeding day (day 0) are set at 1.0. **p<0.01; significant difference between from mock-transfected 4T1 cells at corresponding time points. ($F_{7, 38}$ = 425.953, two-way ANOVA with the Tukey–Kramer test). (F) Difference in the spheroid formation ability between mock-transduced and CLOCK-expressing 4T1 cells. The left panel shows a representative photograph of the Hoechst-stained spheroids formed by mock-transduced or CLOCK-expressing 4T1 cells. The right panel shows the number of spheroid and the parcellation of the diameter. Values show the mean with SD (n = 3). **p<0.01; significant difference from mock-transduced 4T1 cells ($t_4$ = 27.067, Student's t-test, for number of spheroids).

The online version of this article includes the following source data and figure supplement(s) for figure 2:

**Source data 1.** This spreadsheet contains the source for *Figure 2*.

**Figure supplement 1.** CLOCK regulates the expression of *Aldh3a1* through mediation of C/EBPα.

**Figure supplement 1—source data 1.** This spreadsheet contains the source for *Figure 2—figure supplement 1*.

---

enhanced CLOCK-expressing 4T1 cells (*Figure 3C* and *Figure 3D*). In contrast, the expression of mesenchymal molecules, Vimentin, Snail1, and Twist1, was decreased in CLOCK-expressing 4T1 cells (*Figure 3C* and *Figure 3D*). Taken together, the in vitro data suggest attenuation of 4T1 cells invasive potential through EMT suppression by enhancement of CLOCK expression.

## Anti-tumor effects of CLOCK in 4T1 cells-bearing mice

Next, we investigated the anti-tumor effects of CLOCK in 4T1 cell-bearing mice. Mock-transduced or enhanced CLOCK-expressing 4T1 cells were implanted into the mammary fat pad of female BALB/c mice, and the tumor volume was measured every week. The growth rate of enhanced CLOCK-expressing 4T1 cell-formed tumors was significantly lower than that of tumors formed by Mock-transduced 4T1 cells (p<0.01, *Figure 4A*). Similarly, low intensity of immunohistological staining for Ki-67 was observed in enhanced CLOCK-expressing 4T1 tumors (p<0.01, *Figure 4B*). Moreover, CLOCK-expressing 4T1 cell-bearing mice had limited formation of tumor colonies in lung and bone marrow (p<0.01, *Figure 4C and D*). The in vivo results reveal suppression of tumor malignancy by enhancement of CLOCK expression in 4T1 cells.

## Post-transcriptional regulation of *Clock* mRNA expression in ALDH-positive 4T1 cells

To investigated the underlying mechanisms of downregulation of *Clock* expression in ALDH-positive 4T1 cells may be a therapeutic target for overcoming malignant properties of BCSCs; hence, we constructed a luciferase reporter containing the 5'-upstream region of the mouse *Clock* gene (*Clock*::Luc). The 5'-upstream regions included ROREs (orphan receptor response elements), RARE (retinoic acid response element), and an E-box like sequence (*Figure 5A* left). The expression levels of *Luciferase* mRNA derived from *Clock*::Luc in ALDH-positive 4T1 cells was slightly, but significantly, higher than in ALDH-negative cells (p<0.01, *Figure 5A* right). Moreover, no significant difference in mRNA levels of transcription factors regulating *Clock* gene expression between ALDH-positive and -negative cells was found (*Supplementary file 2*). This suggests that suppression of *Clock* gene expression in ALDH-positive 4T1 cells was not due to decrease in its transcriptional activity; hence, we changed focus to the post-transcriptional regulation mechanism repressing *Clock* mRNA expression in ALDH-positive cells.

*Luciferase* mRNA level of the reporter vector containing bp +1 to +4522 of the mouse *Clock* 3'UTR (*Clock* 3'UTR Full::Luc) was markedly decreased compared with that of the control pGL4.13 reporter vector without the 3'UTR (*Figure 5B*). Furthermore, the *Luciferase* mRNA level of the *Clock* 3'UTR Full::Luc vector in ALDH-positive 4T1 cells was significantly lower than in ALDH-negative cells (p<0.01), suggesting post-transcriptional regulation of *Clock* expression in ALDH-positive 4T1 cells. Significant differences in the *luciferase* mRNA levels between ALDH-positive and ALDH-negative 4T1 cells were also detected when the cells were transfected with *Clock* 3'UTR 1 st::Luc and *Clock*

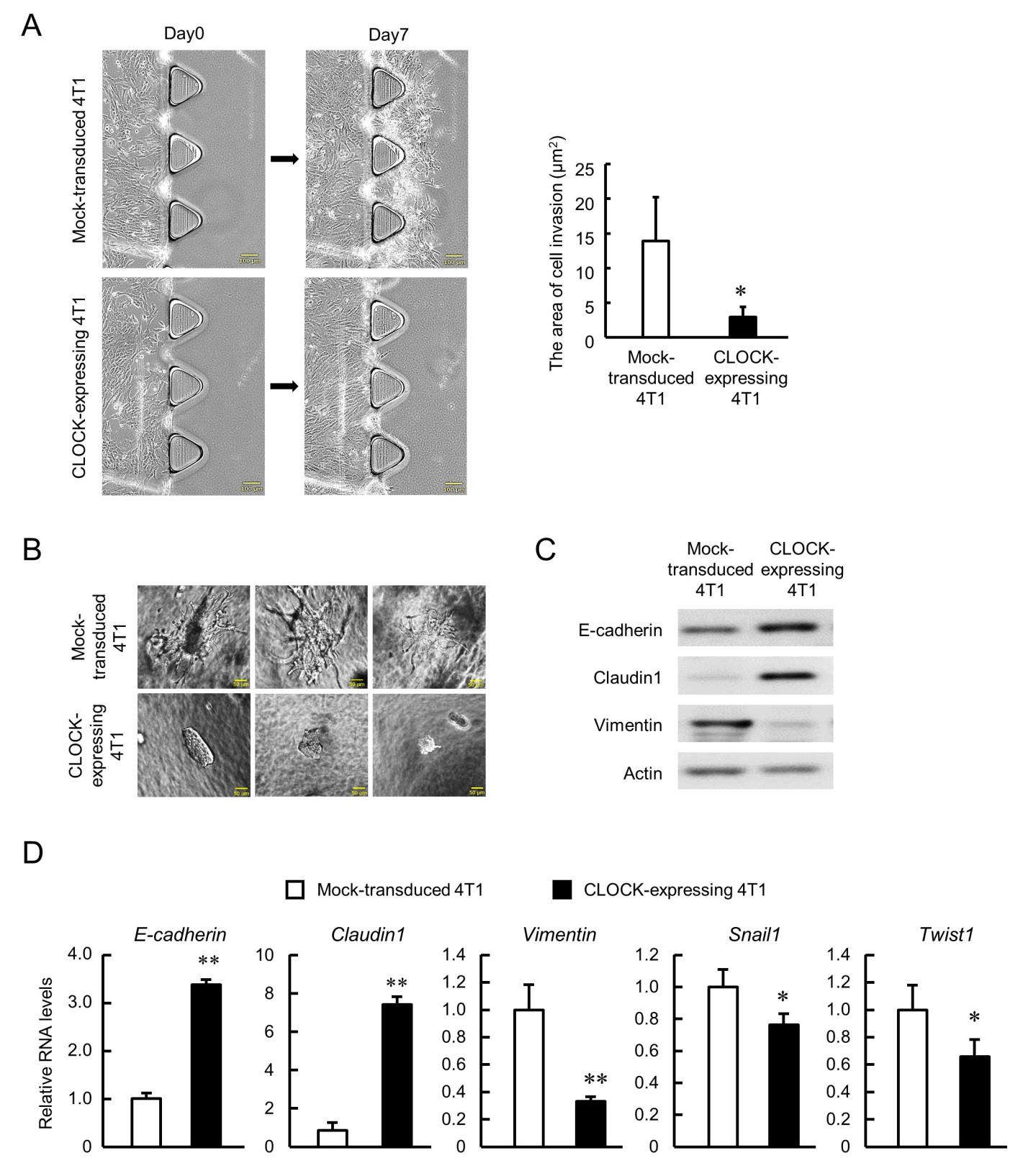

**Figure 3.** Suppression of invasive potential of 4T1 cells by CLOCK. (**A**) Decrease in the invasion ability of CLOCK-expressing 4T1 cells. Microphotographs show invasion of cells into 3D collagen gel. The right panel shows difference in the invasion area between mock-transduced and CLOCK-expressing 4T1 cells. Values show mean with SD (n = 3). *p<0.05; significant difference from mock-transduced 4T1 cells ($t_4$ = 2.968, Student's

*Figure 3 continued on next page*

*Figure 3 continued*

t-test). (**B**) Representative microphotographs of spheroid invasion by mock-transduced and CLOCK-expressing 4T1 cells. Invasive morphology was detected by mock-transduced 4T1 cells. (**C**) Difference in the expression of adhesion molecules between mock-transduced and CLOCK-expressing 4T1 cells. (**D**) Differential expression of EMT-related molecules between mock-transduced and CLOCK-expressing 4T1 cells. *E-cadherin* and *Claudin1* indicate the 'epithelial' state, and *Vimentin*, *Snail1*, and *Twist1* indicate the 'mesenchymal' state. Data were normalized by the *18s rRNA* levels. Values show the mean with SD (n = 3). The values of the mock-transduced group are set at 1.0. \*\*p<0.01, \*p<0.05; significant difference compared with mock-transduced 4T1 cells ($t_4$ = 28.550 for *E-cadherin*; $t_4$ = 18.159 for *Claudin1*; $t_4$ = 6.233 for *Vimentin*; $t_4$ = 2.873 for *Snail1*; $t_4$ = 2.881 for *Twist1*; Student's t-test).

The online version of this article includes the following source data for figure 3:

**Source data 1.** This spreadsheet contains the source for *Figure 3*.

3'UTR 3rd::Luc (p<0.05 respectively, *Figure 5B*). These results suggest that specific 3'UTRs of mouse *Clock* mRNA are responsible for its post-transcriptional regulation in ALDH-positive 4T1 cells.

## miR-182 suppresses *Clock* mRNA expression in ALDH-positive 4T1 cells

miRNAs suppress target gene expression by targeting the mRNA 3'UTR (*Lewis et al., 2005*). Therefore, we attempted to identify the miRNA repressing the expression of *Clock* mRNA in ALDH-positive cells by conducting miRNA microarray analysis. The analysis was done by setting three identification criteria: (*Acharyya et al., 2012*) miRNA expression twofold higher in ALDH-positive cells than in ALDH-negative cells, with signal intensity of at least over 100; (*Al-Hajj et al., 2003*) miRNAs binding to the *Clock* 3'UTR based on the miRDB database (target score >90); and (*Anderson et al., 2011*) miRNAs binding to the *Clock* 3'UTR from bp +1 to +900 or from bp +1922 to +3600 based on the microRNA.org database (*Figure 6A*). These analyses identified mmu-miR-182 as the miRNA upregulated in ALDH-positive cells with the greatest fold change (*Supplementary file 3*). There was no significant difference in the levels of previously known *Clock* gene-targeting miRNAs like miR-1306, miR-290–295 family, miR-17 (*Umemura et al., 2017*), and miR-211 (*Bu et al., 2018*). Therefore, we focused on miR-182 as a possible candidate to repress *Clock* mRNA expression in ALDH-positive cells.

The expression of miR-182 in ALDH-positive cells was significantly higher than that in ALDH-negative cells (p<0.01, *Figure 6B*). Treatment with a miR-182 inhibitor increased the luciferase activity in *Clock* 3'UTR 1 st::Luc-transfected 4T1 cells (*Figure 6—figure supplement 1A*); indicating post-transcriptional regulation of mouse *Clock* mRNA expression through its 3'UTR from bp +1 to +900 by miR-182. We prepared miR-182 knockout (KO) 4T1 cells using CRISPR/Cas9 system (*Figure 6—figure supplement 1B*). Three clones were isolated and deletion of genomic region of miR-182 was confirmed (*Figure 6—figure supplement 1C*). *Clock* gene mRNA levels significantly increased in the ALDH-positive population of all three miR-182 KO 4T1 cells (*Figure 6C*), indicating that miR-182 suppresses the expression of *Clock* mRNA in ALDH-positive 4T1 cells.

## Depletion of miR-182 suppresses tumor growth in 4T1 cells-bearing mice

The final set of experiments investigated whether depletion of miR-182 affects tumor malignant properties. Naive 4T1 or miR-182 KO 4T1 #1 cells were implanted into the mammary fat pad of female BALB/c mice. The tumor growth of miR-182 KO 4T1 cell-bearing mice was significantly lower than that of tumors formed by naive 4T1 cells (p<0.01, *Figure 7A*). The expression levels of *Clock* mRNA in resected tumor were significantly increased by knockout of miR-182 (*Figure 7—figure supplement 1A*), and the effect was more potent in ALDH-positive cell (*Figure 7—figure supplement 1B*). Limited number of tumor colonies were observed in miR-182 KO cells implanted-mice lung (p<0.01, *Figure 7B*), also in bone marrow (p<0.01, *Figure 7C*). These data suggest that tumor malignancy can be attenuated by targeting miR-182 through post-transcriptional activation of CLOCK. Furthermore, highly expression levels of miR-182 were detected in naive 4T1 cell-formed tumor compared with other organs (*Figure 7D*). Similar results were also confirmed in clinical data from miR-TV database (*Figure 7—figure supplement 2*), suggesting that tumor-selective effects were expected by targeting miR-182. Taken together, miR-182-mediated post-transcriptional regulation of CLOCK may be a novel insight for treatment of breast cancer attenuating tumor malignancy.

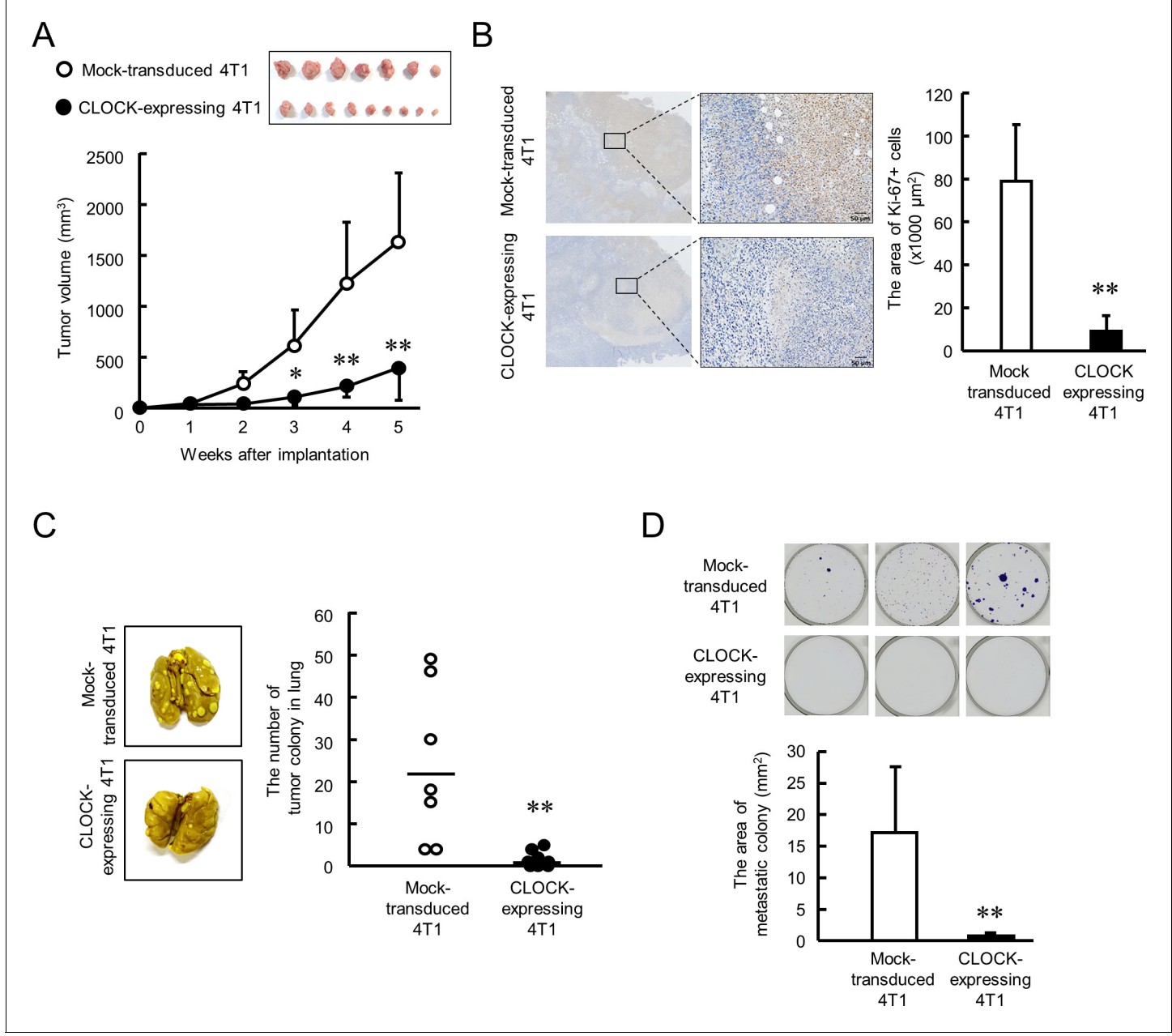

**Figure 4.** CLOCK-induced suppression of malignancy of 4T1 tumors in mice. (**A**) The ability of tumor growth in mice implanted with mock-transduced or CLOCK-expressing 4T1 cells. Top panel shows the photograph of tumor burden isolated from each 4T1 cell implanted mice 6 weeks after implantation. Values are the mean with SD (n = 8–9 animals). **p<0.01; *p<0.05, significant difference compared with mock-transduced 4T1 cell-implanted mice at corresponding time points ($F_{9, 67}$ = 19.956, p<0.001; two-way ANOVA with the Tukey–Kramer test). (**B**) Immunohistochemical staining of Ki-67 in mock-transduced or CLOCK-expressing 4T1 tumors. Complexes with Ki-67 and antibodies were visualized by 3, 3'-diaminobenzidine (brown), and nuclei were stained with hematoxylin (blue). Scale bars indicate 50 μm. Values show mean with SD (n = 6 animals). **p<0.01; significant difference from mock-transduced 4T1 cells ($t_{10}$ = 6.213, Student's t-test). (**C**) The number of pulmonary tumor colonies in mice implanted with mock-transduced or CLOCK-expressing 4T1 cells. Pulmonary colonies were assessed 6 weeks after implantation. The left panels show representative photographs of pulmonary tumor colonies of mock-transduced or CLOCK-expressing 4T1 cells implanted in mice. Right panel shows the quantification of the number of tumor colonies in lungs (n = 7–9 animals). **p<0.01; significant difference from mock-transdaced 4T1 ($t_{14}$ = 3.601; Student's t-test). (**D**) The number of metastatic colonies isolated from tumor-bearing mice femora bone marrow. The top panels show the representative photograph of tumor colonies stained by crystalbiolet. The bottom panel shows the quantification of the area of tumor colonies. Values show mean with SD (n = 8 animals). **p<0.01; significant difference from mock-transduced 4T1 cells ($t_{14}$ = 3.871, Student's t-test).

The online version of this article includes the following source data for figure 4:

**Source data 1.** This spreadsheet contains the source for *Figure 4*.

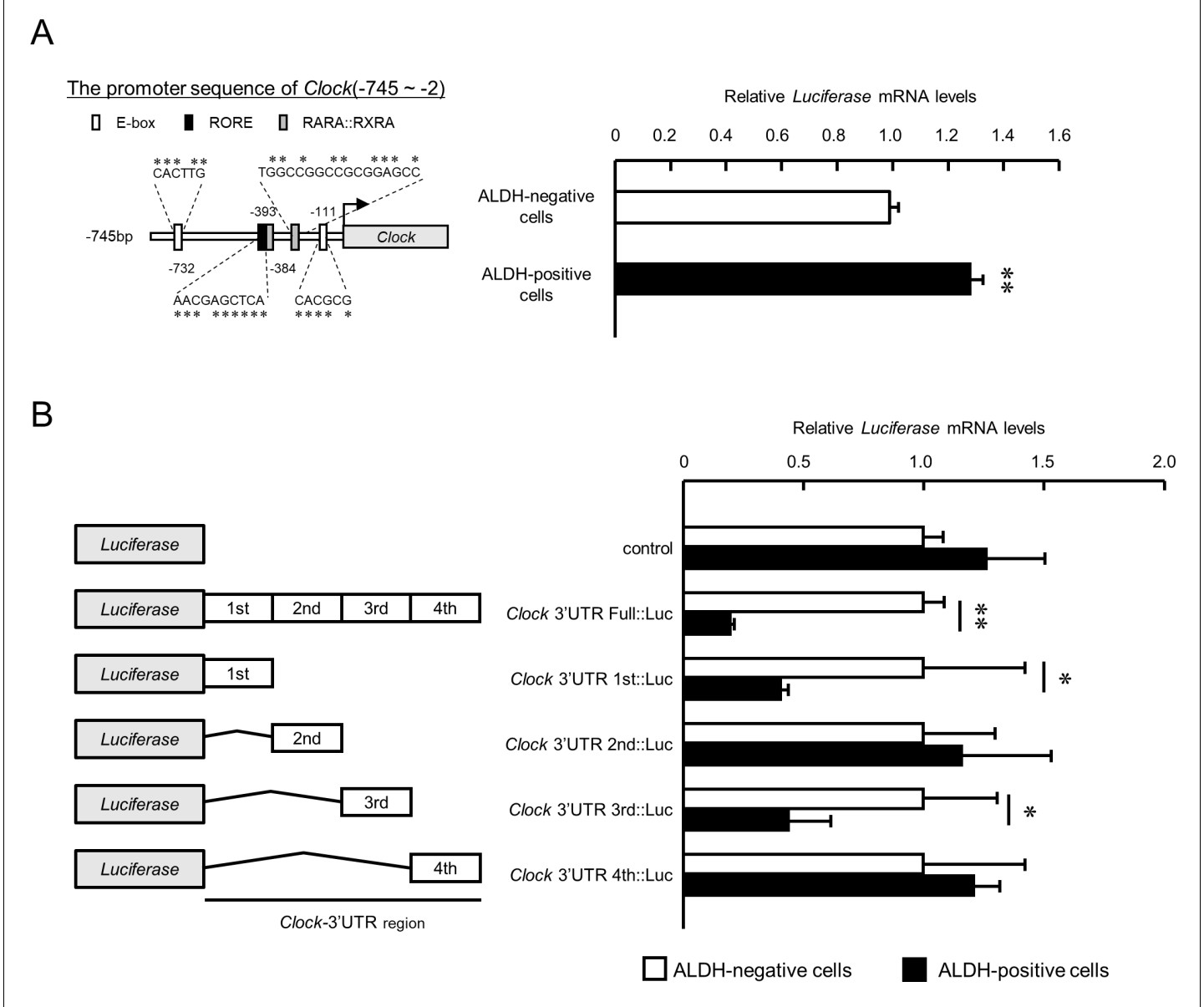

**Figure 5.** The post-transcriptional regulation of *Clock* gene expression in ALDH-positive 4T1 cells. (**A**) Comparison of the promoter activity of the mouse *Clock* gene between ALDH-negative and -positive 4T1 cells. Left panel shows the schematic representation of the mouse *Clock* gene promoter. Numbers near the boxes are nucleotide residues in which E-box, RORE, and RAR/RXR response elements are positioned relative to the transcription start site (+1). Right panel shows the mRNA expression of the *Luciferase* gene *Clock*::Luc in ALDH-negative and -positive 4T1 cells. Data were normalized by *Neomycin phosphotransferase* (*Neo'*) as an internal control. Values are the mean with SD (n = 3).**p<0.01;significant difference between from ALDH-negative cells. The value of ALDH-negative cells is set at 1.0. (**B**) The mRNA expression of the *Luciferase* gene in ALDH-negative and -positive 4T1 cells transfected with luciferase reporter constructs containing varying lengths of the 3'UTR of the *Clock* gene. Data were normalized by the expression levels of *hRLuc* mRNA. Values are the mean with SD (n = 3). *p<0.05, **p<0.01; significant difference between the two groups ($t_4$ = 7.665 for *Clock*-3'UTR Full::Luc; $t_4$ = 4.375 for *Clock*-3'UTR 1 st::Luc; $t_4$ = 2.751 for *Clock*-3'UTR 3rd:: Luc; Student's t-test).

The online version of this article includes the following source data for figure 5:

**Source data 1.** This spreadsheet contains the source for *Figure 5*.

## Discussion

The dysfunction of circadian clock genes is associated with tumor malignancy (*Katamune et al., 2019*; *Katamune et al., 2016*; *Masri et al., 2015*); hence, activation of clock function can be a

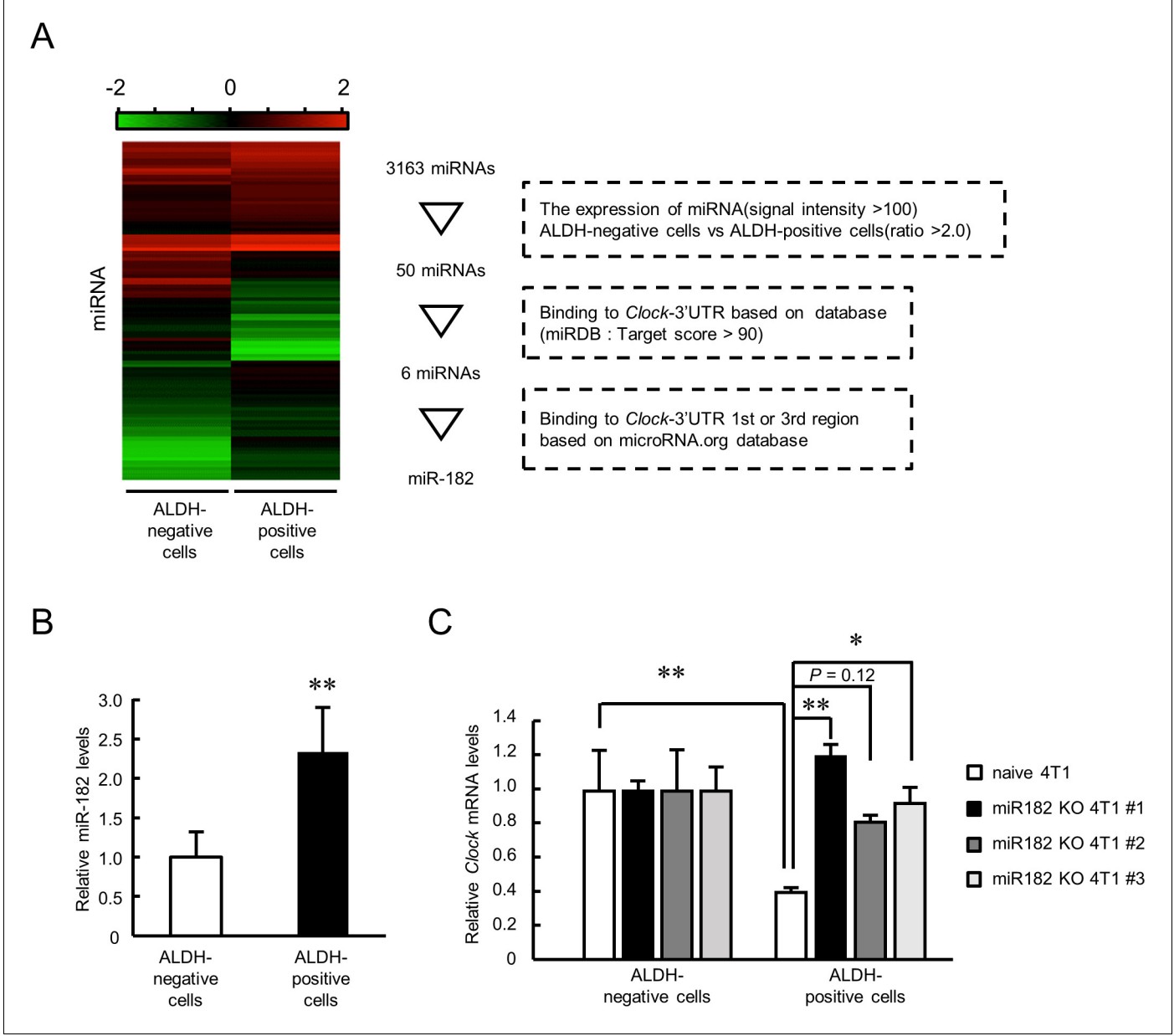

**Figure 6.** Role of miR-182 in post-transcriptional regulation of *Clock* gene expression in ALDH-positive 4T1 cells. (**A**) Procedure of selecting miRNA that regulates *Clock* expression in ALDH-positive 4T1 cells. Heatmap shows the differential expression of miRNA between ALDH-negative and -positive 4T1 cells. microRNA target prediction databases (miRDB and microRNA.org) were applied to this selection. (**B**) The expression levels of mmu-miR-182 in ALDH-negative and -positive cells. Data were normalized by the *β-Actin* mRNA levels. Values are the mean with SD (n = 5). The value of ALDH-negative cells is set at 1.0. \*\*p<0.01; significant difference from ALDH-negative cells ($t_8$ = 3.828, Student's t-test). (**C**) miR-182 negatively regulates the expression of *Clock* mRNA in ALDH-positive 4T1 cells. Three miR-182 KO clones were selected for this experiment. Data were normalized by the *18 s rRNA* levels. Values are the mean with SD (n = 3). The values of ALDH-negative cells are set at 1.0. \*p<0.05, \*\*p<0.01; significant difference between two groups ($F_{7,16}$ = 7.681, p<0.01, ANOVA with the Tukey–Kramer post hoc test).

The online version of this article includes the following source data and figure supplement(s) for figure 6:

**Source data 1.** This spreadsheet contains the source for *Figure 6*.

**Figure supplement 1.** Influence of functional depletion of miR-182 on the regulation of *Clock* expression.

**Figure supplement 1—source data 1.** This spreadsheet contains the source for *Figure 6—figure supplement 1*.

**Figure supplement 2.** The influence of genomic depletion of miR-182 on the other clustered miRNAs.

**Figure supplement 2—source data 1.** This spreadsheet contains the source for *Figure 6—figure supplement 2*.

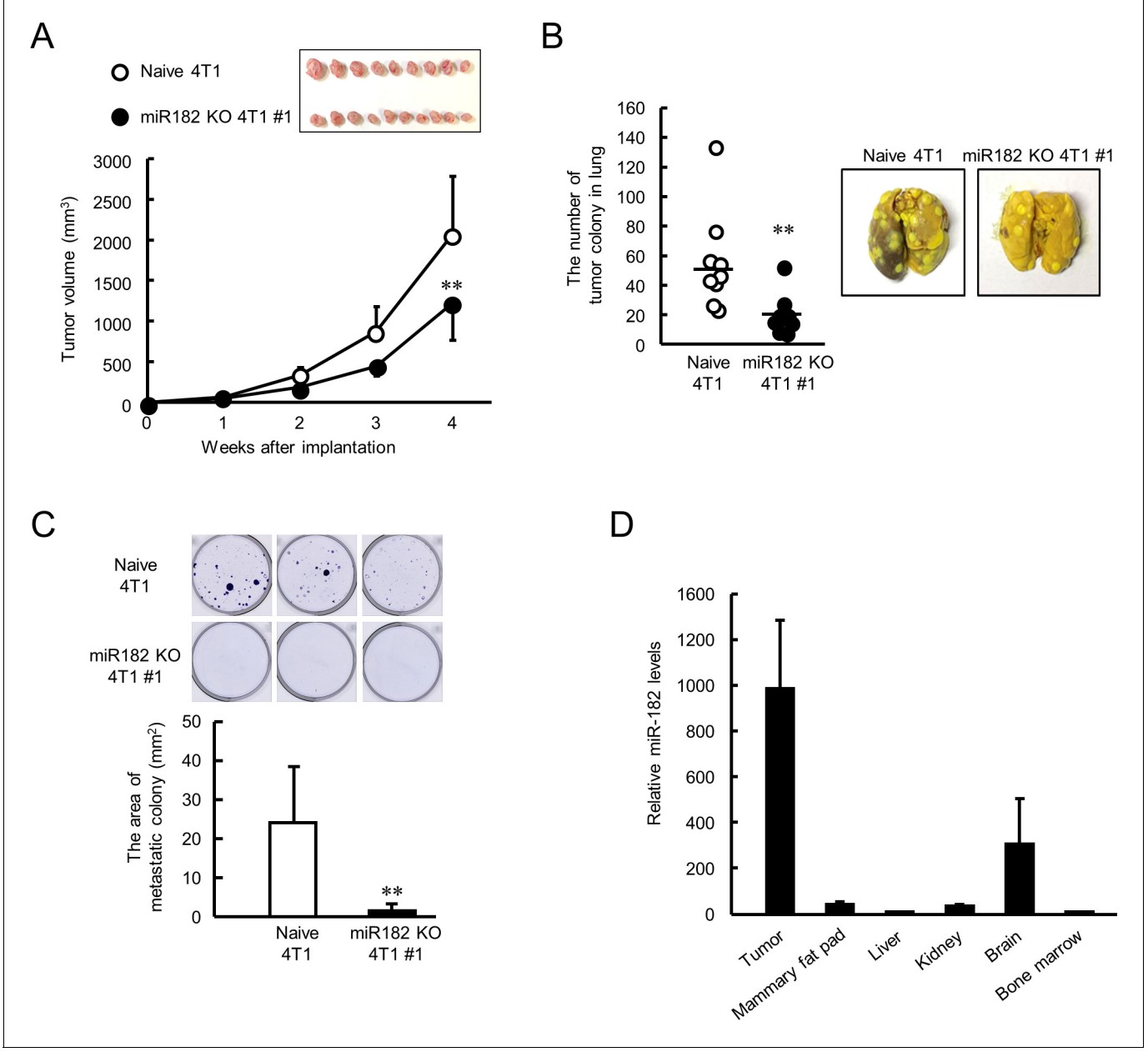

**Figure 7.** Suppression of malignant properties of 4T1 tumor by knockout of miR-182. (A) The ability of tumor growth in mice implanted with naive or miR-182 KO 4T1 cells. Values are the mean with SD (n = 9–10 animals). **p<0.01; significant difference compared with naive 4T1 cell-implanted mice at corresponding time points ($F_{7, 63}$ = 39.494, p<0.001; two-way ANOVA with the Tukey–Kramer test). Top panel shows the photograph of tumors isolated from each 4T1 cell implanted mice 5 weeks after implantation. (B) The number of pulmonary tumor colonies in mice implanted with naive or miR-182 KO 4T1 cells. Left panel shows the quantification of the number of tumor colonies in lungs (n = 9 animals). **p<0.01; significant difference from naive 4T1 group($t_{16}$ = 2.993; Student's t-test). The right panels show representative photographs of pulmonary tumor colonies of naive or miR-182 KO 4T1 cells implanted in mice. Pulmonary tumor colonies were assessed 5 weeks after implantation. (C) The number of metastatic colonies isolated from tumor-bearing mice femora bone marrow. The top panels show the representative photograph of tumor colonies stained by crystalbiolet. The bottom panel shows the quantification of the area of tumor colonies. Values show mean with SD (n = 6 animals). **p<0.01; significant difference from naive 4T1 cells ($t_{10}$ = 3.354, Student's t-test). (D) The organ distribution of miR-182 in naive 4T1-bearing mice. Data were normalized by the *18 s rRNA* levels. Values are the mean with SD (n = 3 animals). The value of bone marrow is set at 1.0.

The online version of this article includes the following source data and figure supplement(s) for figure 7:

**Source data 1.** This spreadsheet contains the source for *Figure 7*.

**Figure supplement 1.** The expression of CLOCK in miR-182 KO 4T1 cell #1-bearing mice tumor.

**Figure supplement 1—source data 1.** This spreadsheet contains the source for *Figure 7—figure supplement 1*.

*Figure 7 continued on next page*

**Figure supplement 2.** The expression levels of hsa-miR-182 in breast tumor and normal mammary gland derived from patients.
**Figure supplement 3.** Sequence conservation of miR-182.
**Figure supplement 4.** Database analysis of clinical relevance of CLOCK expression to patients' outcome.

therapeutic strategy for treatment of cancers. Our current study demonstrates a relationship between the malignancy of BCSCs and the expression of CLOCK, a major component of the molecular circadian machinery. Low expression levels of CLOCK were detected in ALDH-positive 4T1 cells, and enhancing CLOCK expression in 4T1 cells attenuated tumor growth and invasive potential. Furthermore, the expression of CLOCK in ALDH-positive 4T1 cells is regulated at the post-transcriptional level and may be a target to attenuate the malignancy of 4T1 cells (*Figure 8*).

Enhancement of CLOCK expression decreased the number of ALDH-positive populations of 4T1 cells. The detoxification capacity of ALDH can protect stem cells against oxidative damage and is an important factor in their longevity (*Bigarella et al., 2014*; *Ito et al., 2006*). High ALDH activity also protects cells from the cytotoxic effects of chemotherapeutic drugs through the degradation of reactive oxygen species (*Awad et al., 2010*; *Katamune et al., 2019*). High activity of ALDH correlates with tumor grade, metastasis, and poor prognosis in patient breast tumor (*Charafe-Jauffret et al., 2010*; *Marcato et al., 2011*); consequently, low expression levels of CLOCK in ALDH-positive 4T1 cells likely promote their malignant properties by regulating ALDH activity. C/EBPα was identified as a mediator of CLOCK-induced ALDH suppression. Reports suggest that upregulation of C/EBPα contributes suppression of breast tumors through induction of apoptosis and inhibition of cell

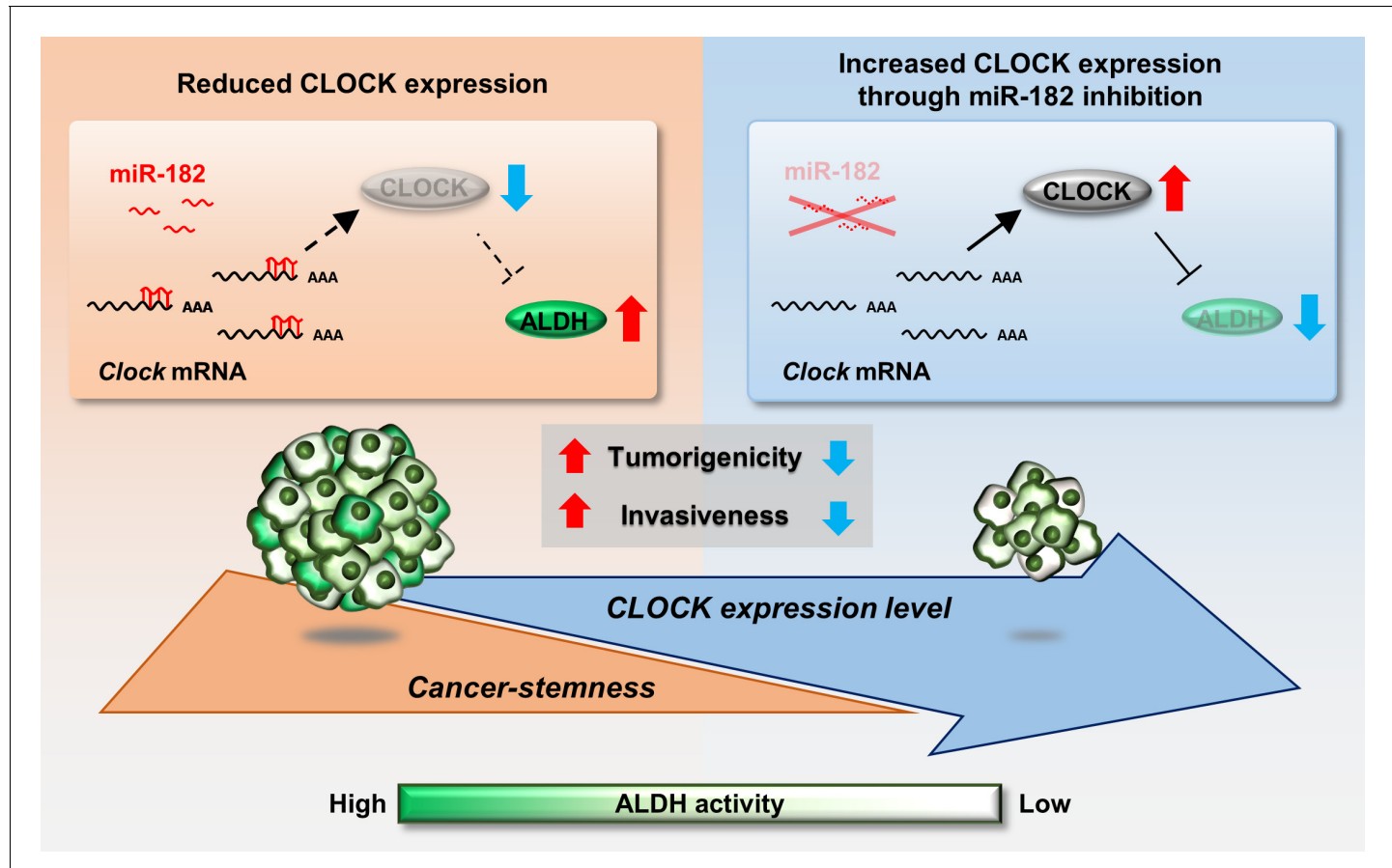

**Figure 8.** Schematic diagrams indicating the role of CLOCK in the regulation of malignancy of breast cancer stem-like cells. mmu-miR-182 post-transcriptionally suppresses the expression of *Clock* in ALDH-positive 4T1 cells, leading to maintenance of their malignancy. Increased CLOCK levels attenuate the stemness properties of ALDH-positive cells.

proliferation (*Lourenço and Coffer, 2017*). These tumor-suppressive effects may also correlate with repressing the transcription of *Aldh3a1*. However, further investigation is required to evaluate whether this CLOCK-induced ALDH suppression can be generalized to all types of cancer cells because the function of C/EBPα is altered by organs, differentiation state of cells, and transcriptional co-regulators (*Lourenço and Coffer, 2017*; *Ossipow et al., 1993*; *Ramji and Foka, 2002*).

The circadian clock genes are ubiquitously expressed throughout the body, generating circadian oscillations of numerous biological processes at the molecular level (*Yoo et al., 2004*; *Zvonic et al., 2006*). The molecular oscillations underlying the generation of circadian rhythms develop gradually during ontogenesis. Oscillation in the expression of clock genes is not detected in immature cells such as mouse embryonic stem (ES) cells and induced pluripotent stem (iPS) cells (*Yagita et al., 2010*). In particular, the expression of CLOCK is decreased in mouse embryos, and increasing CLOCK expression during development is considered to be important for normal differentiation (*Umemura et al., 2017*). We previously reported that oscillations of clock genes were dampened in ALDH-positive 4T1 cells compared with those in ALDH-negative cell populations (*Matsunaga et al., 2018*). In this study, we also observed a decrease in expression of CLOCK mRNA and protein in ALDH-positive 4T1 cells in vivo and in vitro, with lower expression decreasing E-box-mediated transcriptional activity. Increased CLOCK expression by lentiviral transduction enhanced oscillation of E-box-mediated reporter luciferase bioluminescence in ALDH-positive 4T1 cells. It is difficult to distinguish the role of CLOCK in the regulation of ALDH expression from its role in the molecular circadian machinery; however, dampened rhythm of clock gene expression in ALDH-positive 4T1 cells may contribute to their stem-like properties. Further studies are required to investigate the relationship between enhanced oscillation of circadian gene expression and attenuation of cancer malignancy.

The malignancy of CSCs is altered depending on their differential states (*Bisson and Prowse, 2009*; *Campos et al., 2010*; *Xiong et al., 2017*; *Tampaki et al., 2017*). The malignant properties of 4T1 cells were attenuated by transduction with CLOCK expressing lentivirus, resulting in decreased tumor growth of tumor-bearing mice. Some pluripotent markers, such as *Klf4*, *Nanog*, and *Myc*, have important roles in maintenance of the undifferentiated state of cancer cells (*Jeter et al., 2011*; *Lathia and Liu, 2017*; *Lawson et al., 2015*; *Yu et al., 2011*), negatively correlated with treatment outcomes in cancer patients (*Cheng et al., 2018*; *Dong et al., 2014*; *Elbadawy et al., 2019*; *Rasti et al., 2018*). In addition to pluripotent marker genes, enhanced expression of CLOCK in 4T1 cells altered the expression of genes to suppress EMT; this may explain the underlying mechanism of attenuation of invasive potential in enhanced CLOCK-expressing 4T1 cells. Cancer cells undergo EMT in response to different environmental conditions, causing a functional and phenotypical transition of polarized epithelial cells into mesenchymal cells. Mesenchymal transcriptional factors such as *Snail1* and *Twist1* promote CSC-like phenotypes, resulting in resistance to treatment, cancer recurrence, and metastasis (*Liang et al., 2015*; *Mani et al., 2008*; *Ren et al., 2016*). Increased levels of *Nanog* can also generate mesenchymal-like cells by repressing the expression of *E-cadherin* (*Siu et al., 2013*). The prevention of EMT by altered gene expression in enhanced CLOCK-expressing 4T1 cells may contribute to the attenuation of CSC-like properties. 4T1 cells are reported metastasize to lung, liver, brain, and bone (*Pulaski and Ostrand-Rosenberg, 2000*). 4T1 is lung tropic cancer cells (*Monteran et al., 2020*); thus, metastatic colonies were detected mainly from lung in this study. Only few tumor colonies were detected except for bone at this stage. To evaluate the effect of CLOCK expression on metastatic capacities to multiple organs, further investigations are also needed to employ several cancer cells.

Reporter gene analysis revealed that the downregulation of CLOCK in ALDH-positive 4T1 cells is caused by post-transcriptional suppression through binding of miR-182 to the 3'UTR of *Clock* mRNA. miR-182 is associated with organ development, T cell differentiation, and cancer (*Stittrich et al., 2010*; *Wei et al., 2015*; *Xu et al., 2007*), and belongs to a polycistronic miRNA cluster with miR-96 and miR-183 located within a 4-kbp area in the murine genome (*Wei et al., 2015*). The sequence of miR-96 and miR-183 in the cluster was not affected by genomic deletion of miR-182, but their expression levels were compensatory increased in miR-182 KO 4T1 cell #1 (*Figure 6— figure supplement 2*). Despite an increase in miR-96 and miR-183 levels, the deletion of miR-182 increased the expression of CLOCK and decreased tumor growth. Hence, miR-182 plays a major role in tumor biology and the post-transcriptional regulation of CLOCK. The polycistronic cluster of miR-96/miR-182/miR-183 is highly conserved in sequence and functions between humans and mice,

and also the 3'UTR sequence of human *CLOCK* gene around miR-182 binding site is highly conserved (*Figure 7—figure supplement 3*). High miR-182 levels in human breast tumor tissue correlate with poor prognosis (*Chiang et al., 2013*; *Guttilla and White, 2009*; *Lei et al., 2014*), which suggest clinical importance of miR-182 for cancer patient's prognosis.

Patients with early breast cancers are considered to have a good prognosis; however, aggressive breast cancers, such as TNBCs, are more serious (*Dent et al., 2007*; *Luo et al., 2013*). BCSCs are possible therapeutic targets for better prognosis, but standard treatments targeting BCSCs are not available. The tumor microenvironment may be responsible for several malignant properties like recurrence, metastasis, and resistance to treatments (*Acharyya et al., 2012*; *Meads et al., 2009*; *Shah et al., 2015*), and studies reveal that these malignant properties are caused at the molecular level, with the interaction between BCSCs and the microenvironment also associated with tumor malignancy. Therefore, efforts to develop novel therapy targeting molecules responsible for the interaction between BCSCs and the microenvironment are underway. However, the complicated molecular interactions make drug development targeting BCSCs difficult; hence, a comprehensive understanding of BCSC biology is crucial for improvements in breast cancer therapy. The present study provides novel insights to attenuate cancer stemness by regulating the expression of circadian clock genes. The amino acid sequence and function of CLOCK are highly conserved among species; thus, similar effects can be expected in human malignant breast cancers. However, both tumor-promotional and tumor-suppressive relevance are existing between CLOCK expression and outcome (*Figure 7—figure supplement 4*). To assess the true relevance to patient prognosis, evaluation of CLOCK expression levels in CSCs may be needed.

Existing agents increasing the expression of CLOCK may be ineffective for the treatment of CSCs as they increase the levels of CLOCK protein through transcriptional activation (*Solt et al., 2012*; *Sulli et al., 2018*). Antisense oligonucleotides can be a powerful tool for targeting miRNAs, with high affinity to their target (*Scharner and Aznarez, 2021*). Several issues still exist for the development of these antisense therapeutics because of their pharmacokinetics and stabilities. Researches about novel drug delivery carriers and modified nucleotides is developing in this few decades to overcome these weaknesses (*Saarbach et al., 2019*; *Vandghanooni et al., 2020*). Using these technologies, selective targeting of the CLOCK 3'UTR as a novel treatment strategy for malignant breast cancers is needed in order to therapeutically activate the circadian clock genes in CSCs.

## Materials and methods

**Key resources table**

| Reagent type (species) or resource | Designation | Source or reference | Identifiers | Additional information |
|---|---|---|---|---|
| Gene (*Mus musculus*) | *Clock* | NCBI Gene Database | NCBI Gene: 12753 | |
| Cell line (*Mus musculus*) | 4T1 | ATCC | Cat #: CRL-2539; RRID:CVCL_0125 | |
| Cell line (*Homo sapiens*) | Lenti-X 293T | Clonetech | Cat #: 632180 | |
| Transfected construct (*Mus musculus*) | miRDIAN Hairpin Inhibitor | Horizon Discovery | Cat #: IH-310436–08 | |
| Antibody | Rabbit polyclonal anti-CLOCK | Abcam | Cat #: ab3517 RRID:AB_303866 | WB (1:4000) |
| Antibody | Rabbit polyclonal anti-p84 (THOC1) | Proteintech | Cat #: 10920–1-AP RRID:AB_2202239 | WB (1:1500) |
| Antibody | Goat polyclonal anti-E-Cadherin | R and D systems | Cat #: AF748 RRID:AB_355568 | WB (1:1000) |
| Antibody | Rabbit polyclonal anti-Claudin 1 | Proteintech | Cat #: 13050–1-AP RRID:AB_2079881 | WB (1:2000) |

*Continued on next page*

*Continued*

| Reagent type (species) or resource | Designation | Source or reference | Identifiers | Additional information |
|---|---|---|---|---|
| Antibody | Mouse monoclonal anti-Vimentin | R and D systems | Cat #: MAB21052 RRID:AB_2832972 | WB (1:1000) |
| Antibody | Goat polyclonal anti-Actin-HRP | Santa Cruz Biotechnology | Cat #: sc-1616 | WB (1:5000) |
| Antibody | Mouse monoclonal anti-Ki-67 | Agilent Technology | Cat #: M7240 RRID:AB_2142367 | IHC (1:200) |
| Recombinant DNA reagent | pcDNA3.1(+) (plasmid) | Invitrogen | Cat #: V79020 | |
| Recombinant DNA reagent | pGL4.18 (plasmid) | Promega | Cat #: E6731 | |
| Recombinant DNA reagent | pGL4.13 (plasmid) | Promega | Cat #: E6681 | |
| Recombinant DNA reagent | pRL-SV40 (plasmid) | Promega | Cat #: E2231 | |
| Sequence-based reagent | Primer for sgRNA synthesis | This paper | | CCTCTAATA CGACTCACT ATAGGCAAT GGTAGAACT CACACGTTT AAGAGCTAT GC |
| Sequence-based reagent | Mouse *Clock* promoter F | This paper | PCR primers for construction of reporter vector | ATACTCGAGA GGTCACTTG GGTCGT |
| Sequence-based reagent | Mouse *Clock* promoter R | This paper | PCR primers for construction of reporter vector | ATAAGATCT CCTTCCCCT CCTCCACG |
| Peptide, recombinant protein | Recombinant Cas9 | Clontech | Cat #: Z2641N | |
| Peptide, recombinant protein | Recombinant Mouse TGF-β1 | R and D systems | Cat #: 7666 MB | |
| Commercial assay or kit | ALDEFLUOR Kit | StemCell Technologies | Cat #: ST-01700 | |
| Commercial assay or kit | Lipofectamine LTX and Plus Regent | ThermoFisher SCIENTIFIC | Cat #: 15338100 | |
| Commercial assay or kit | Dual-Luciferase reporter assay system | Promega | Cat #: E1910 | |
| Commercial assay or kit | ReverTra Ace qPCR RT Kit | Toyobo | Cat #: FSQ-201 | |
| Commercial assay or kit | THUNDERBIRD SYBR qPCR Mix | Toyobo | Cat #: QPS-201 | |
| Commercial assay or kit | Taqman MicroRNA Reverse Transcription Kit | Applied Biosystems | Cat #: 4366596 | |
| Commercial assay or kit | Taqman MicroRNA Assay | Applied Biosystems | Cat #: 4427975 | |
| Commercial assay or kit | Guide-it sgRNA in vitro transcription kit | Clontech | Cat #: Z2635N | |
| Commercial assay or kit | Lentiviral High Titer Packaging Mix with pLVSIN series | Takara bio | Cat #: 6952 | |

*Continued on next page*

*Continued*

| Reagent type (species) or resource | Designation | Source or reference | Identifiers | Additional information |
|---|---|---|---|---|
| Commercial assay or kit | CellTiter-Glo Luminescent cell viability assay | Promega | Cat #: 7572 | |
| Chemical compound, drug | 6-Thioguanine | Fujifilm | Cat #: 203–03771 | |
| Software, algorithm | BD FACS Diva | BD Biosciences | RRID:SCR_001456 | |
| Software, algorithm | FlowJo | BD Biosciences | RRID:SCR_008520 | |
| Software, algorithm | Lumicycle analysis software | Actimetrics | | |
| Software, algorithm | ImageQuant LAS 3000 | GE Healthcare | RRID:SCR_014246 | |
| Software, algorithm | BZ analyzer software | KEYENCE | RRID:SCR_017205 | |
| Software, algorithm | Off-Spotter | Thomas Jefferson University | RRID:SCR_015739 | |
| Software, algorithm | JMP | Statistical Discovery | RRID:SCR_014242 | |
| Other | Hoechst33342 | Dojindo laboratories | Cat #: 346–07951 | |

## Cells and treatments

4T1 mouse breast cancer cells were purchased from American Type Culture Collection, and cultured under a 5% $CO_2$ environment at 37°C in Roswell Park Memorial Institute (RPMI)−1640 medium supplemented with 10% FBS and 1% penicillin/streptomycin on two-dimensional (2D) CELLSTAR cell culture flasks (Greiner Bio-One, Monroe, NC) or synthetic three-dimensional (3D) scaffold biomaterials, Vecell 3D-inserts (Vecell Inc, Kitakyusyu, Japan). We confirmed that there was no microbial in this cell line using a TaKaRa PCR Mycoplasma Detection Set. We confirmed that cell lines were authenticated by ATCC using short tandem repeat (STR) PCR analysis, and these cell lines were used in less than 3 months from frozen stocks.

## Construction of expression and reporter vectors

Expression vectors for mouse BMAL1, CLOCK, CRYPTOCHROME1 (CRY1), PERIOD1 (PER1), PERIOD2 (PER2), and C/EBPα were constructed using cDNA generated from mouse liver RNA by RT-PCR. The coding regions were ligated into the pcDNA3.1(+) vector (Invitrogen; Life Technologies, Carlsbad, CA). MicroRNA expression vectors against mouse *Cebpα* gene were constructed using BLOCK-IT Pol II miR RNAi Expression Vector kit (Invitrogen), according to the manufacturer's instruction. In order to construct reporter vectors, the mouse *Clock* gene promoter region spanning from −745 to −8 bp (the distance in base pairs from the putative transcription start site, +1) was also amplified by PCR, and the product was ligated into the pGL4.18 luciferase reporter vector (Promega, Madison, WI). The primer sequences used for amplification of the mouse *Clock* gene promoter region were as follows: forward primer, 5'-ATACTCGAGAGGTCACTTGGGTCGT-3'; reverse primer, 5'-ATAAGATCTCCTTCCCCTCCTCCACG-3'. The mouse *Aldh3a1* gene promoter region spanning from −2000 to +15 bp was also amplified by PCR, and the product was ligated into the pGL4.18 luciferase reporter vector (Promega). The primer sequences used for amplification of the mouse *Aldh3a1* gene promoter region were as follows: forward primer, 5'-ATACTCGAGAACCCTGGAGACTTTGTTCT-3'; reverse primer, 5'-AATAGATCTTGGAACTCCTGGAATAAGCAAG-3'.

The pGL4.13 reporter vector in which the mouse *Clock* mRNA 3'-untranslated region (3'UTR) was inserted downstream of the luciferase gene was also constructed. To prepare the reporter constructs

containing varying lengths of the 3'UTR of the mouse *Clock* gene, the nucleotide immediately after the stop codon in exon 23 of the mouse *Clock* gene was defined as +1. The 3'UTR of the mouse *Clock* gene (+1 to +4522) was divided into four sections from bp +1 to +900 (*Clock* 3'UTR 1 st::Luc), from bp +901 to +1921 (*Clock* 3'UTR 2nd::Luc), from bp +1922 to +3600 (*Clock* 3'UTR 3rd::Luc), and from bp +3601 to +4522 (*Clock* 3'UTR 4th::Luc). Each section of the 3'UTR of the mouse *Clock* gene was amplified by PCR and then cloned downstream of the luciferase gene into the pGL4.13 reporter vector.

## Luciferase reporter assay

3 × E-box luciferase reporter vector (3 × E-box::Luc) and pRL-SV40 (Promega) as an internal-control reporter were transfected using lipofectamine LTX reagent according to the manufacturer's instructions. Twenty-four hours after transfection, ALDH-positive (ALDH-high activity) and ALDH-negative (ALDH-low activity) cells were gated based on the ALDEFLUOR assay (StemCell Technologies, Vancouver, Canada) as described below. Collected cells were lysed and then subjected to the Dual-Luciferase reporter assay (Promega). The ratio of firefly to Renilla luciferase activities in each sample served as a measure of normalized luciferase activity.

## ALDEFLUOR assay

Dissociated single cells from cell lines were suspended in ALDEFLUOR assay buffer containing an ALDH substrate, bodipy-aminoacetaldehyde, at 1.5 µM, and incubated for 40 min at 37°C. A specific inhibitor of ALDH, diethylaminobenzaldehyde; DEAB, at a 10-fold molar excess, was used as negative control. Fluorescence-activated cell sorting (FACS; BD Biosciences, San Jose, CA) was performed for more than $1 \times 10^6$ cells under low pressure in the absence of UV light. The data were analyzed by BD FACSDiva software V6.1.3 or FlowJo (BD Biosciences).

## Spheroid formation assay

The ability of cells to grow in an anchorage-independent manner was assessed to evaluate the spheroid formation. Cells were seeded in 10% FBS containing RPMI soft agar at a density of $5 \times 10^3$ cells in 24-well plate. On day 7 after seeding, spheroid formation was assessed by staining with Hoechst33342 (Dojindo Laboratories, Kumamoto, Japan). The number and size of spheroid were measured by BZ analyzer (KEYENCE, Osaka, Japan).

## Real-time bioluminescence tracing

The bioluminescence of cultured 3 × E-box::Luc-expressing 4T1 cells was recorded using a real-time monitoring system (Lumicycle, Actimetrics, Wilmette, USA). The ALDH-positive populations of 3 × E-box::Luc-expressing 4T1 cells were cultured on spheroid forming condition, VECELL 3D inserts with 3D tumorsphere medium XF (PromoCell, Heidelberg, Germany). The 3D inserts were placed in 35 mm dishes and stimulated with 100 nM dexamethasone for synchronization of their circadian clocks. The amplitude of bioluminescence derived from 3 × E-box::Luc was calculated using the Lumicycle analysis software (Actimetrics).

## Quantitative RT-PCR analysis

Total RNA was extracted using RNAiso (Takara Bio Co., Ltd., Shiga, Japan) or ReliaPrep RNA Cell Miniprep System (Promega). cDNA was synthesized using a ReverTra Ace qPCR RT Kit (Toyobo, Osaka, Japan) and amplified by PCR. Real-time PCR analysis was performed on diluted cDNA samples using the THUNDERBIRD SYBR qPCR Mix (Toyobo) with the 7500 Real-time PCR system (Applied Biosystems, Tokyo, Japan). Primer sequences for amplification of target genes are listed in *Supplementary file 1*. For quantitation of miRNA, total RNA was reverse-transcribed by the Taqman MicroRNA Reverse Transcription Kit and Taqman MicroRNA Assay (Applied Biosystems). Data were normalized using *18* s and *β-Actin* mRNAs levels.

## Western blot analysis

Protein samples were prepared by CelLytic MT Cell Lysis Reagent (Sigma-Aldrich St. Louis, MO) supplemented with protease inhibitor cocktail, which contained 2 µg/mL of aprotinin, 2 µg/mL of leupeptin, and 100 µmol/L of phenylmethylsulfonyl fluoride. Then, 20 mg of the protein lysate was

resolved by SDS–PAGE on 8% or 10% gels, transferred to polyvinylidene difluoride membranes, and probed with antibodies against Clock (ab3517; Abcam, Cambridge, UK), E-cadherin (AF748; R and D systems, Minneapolis, MN), Claudin1 (13050–1-AP; proteintech, Rosemont, IL), Vimentin (MAB21052; R and D systems), p84 (THOC1, 10920–1-AP; proteintech), and β-Actin (sc-1616; Santa-Cruz Biotechnology, Texas, TX). Specific antigen–antibody complexes were visualized using horse-radish peroxidase–conjugated secondary antibodies and a chemiluminescence reagent. The membranes were photographed, and the density of each band was analyzed with an ImageQuant LAS 3000 (Fuji Film, Japan).

## Establishment of 4T1 cells stably expressing CLOCK

In order to establish stably CLOCK-expressing 4T1 cells, full-length mouse *Clock* cDNA was subcloned into lentiviral vectors under control of the EF1α promoter. Lentivirus particles were prepared by the Lentiviral High Titer Packaging Mix with pLVSIN series (Clonetech, Palo Alto, CA) with Lenti-X 293 T cell lines. 4T1 cells were infected with *Clock*-expressing lentivirus and maintained in medium containing 5 μg/mL of puromycin.

## Database analysis of *Aldh3a1* promoter region

To identify the repressor of *Aldh3a1* induced by enhanced CLOCK-expressing, 5'-upstream region of the mouse *Aldh3a1* g*ene* was analyzed by JASPAR database. Binding score was set at 95%, and then candidates were narrowed down as follows: (*Acharyya et al., 2012*) candidates registered as transcriptional repressor, (*Al-Hajj et al., 2003*) candidates which have complete E-box element in promoter region and confirmed binding signal of CLOCK in ChIP-Atlas database, and (*Anderson et al., 2011*) candidates confirmed the binding signal in Aldh3a1 promoter region in ChIP-Atlas database.

## Invasion assay

For the collagen invasion assay, cells were seeded on a collagen-filled 3D Cell Culture Chip (AIM BIOTECH, Singapore) according to the manufacturer's instructions. Culture medium containing 20 ng/mL of TGF-β was replaced every day. For the spheroid invasion assay, cells were embedded in vitrogel 3D (TheWell BIOSCIENCE, North Brunswick, NJ) containing 1 mg/mL of collagen type 1 at a density of $2 \times 10^5$ cells/mL. Culture medium was replaced every 2 days. Approximately 1–2 weeks after seeding, cells were observed using the KEYENCE all-in-one microscope BZ-X800.

## Determination of growth rate of cultured cells

Cells were seeded at a density of 500 cells/well in 100 μL of culture medium in 96-well plates. The viability of cells was determined at 24 hr intervals after seeding of cells using ATP luminescent cell visibility assay kit (Promega). Growth rate was calculated dividing the change in cell viability against basal value (day 0).

## Animals and treatments

Five-week-old female Balb/c mice (Charles River Laboratory Japan, Inc; Yokohama, Japan) were housed under a standardized light–dark cycle at 24 ± 1°C and 60 ± 10% humidity with food and water ad libitum. Thirty microliters of medium containing $1 \times 10^5$ of each 4T1 cells were implanted in the mammary fat pads or right hind footpads of mice. The tumor volume was estimated according to the following formula: tumor volume $(mm^3)$ = 0.5 × length (longest diameter) × width (shortest diameter)$^2$. Six weeks after implantation of 4T1 tumor cells into the mice, the lungs were removed, rinsed, and fixed in Bouin's solution to stain the tumor nodules. To isolate metastatic tumor colony in bone marrow, bone marrow cells were collected from mouse femora and treated with RBC lysis buffer (155 mM $NH_4Cl$, 10 mM $NaHCO_3$, 0.1 mM EDTA). Cells were suspended in RPMI medium, and cultured under a 5% $CO_2$ environment at 37°C for 2 weeks. 60 μM of 6-thioguanine (6-TG) were added in culture medium to select 4T1 metastatic tumor cells. Tumor colonies were fixed with methanol, and then stained by 0.2% crystalbiolet for quantification. All experimental procedures were performed after approval and following the guidelines of Kyushu University.

## Immunohistochemical staining of Ki-67

To determine the index of tumor cell proliferation, immunohistochemical staining was performed using monoclonal antibody against Ki-67 (MIB1: M7240; Agilent Technology). Tumor tissues were removed from mice and fixed with 4% paraformaldehyde in PBS at 6 weeks after implantation. Paraffin-embedded tumor sections were deparaffinized and rehydrated through graded ethanol, followed by blocking endogenous peroxidase. Tumor sections subjected to heat-induced antigen retrieval and incubated with primary antibody. Specific antigen–antibody complexes were visualized using horseradish peroxidase–conjugated secondary antibodies with 3,3'-diaminobenzidine (DAB) solution and sections were counterstained with hematoxylin.

## Transcriptional and post-transcriptional assay of *Clock* mRNA

*Clock*::Luc or each *Clock* 3'UTR luciferase constructs was transfected using lipofectamine LTX reagent according to the manufacturer's instructions. Twenty-four hours after transfection, ALDH-positive and ALDH-negative cells were gated based on the ALDEFLUOR assay (StemCell Technologies) as described above. Total RNA was extracted from collected cells, and then *Luciferase* mRNA levels were assessed by qRT-PCR to evaluate transcriptional or post-transcriptional regulation of *Clock* mRNA. To normalize transfection efficiency, the expression levels of *Neomycin phosphotransferase* (*Neo*$^r$) derived from *Clock*::Luc vector was assessed in transcriptional assay. In post-transcriptional assay, the expression levels of *RLuc* mRNA derived from co-transfected pRL-SV40 vector (Promega) for normalization.

## mRNA microarray analysis

ALDH-positive and ALDH-negative 4T1 cells were prepared from cultured 4T1 cell. Total RNA was extracted from cells using a QIAGEN RNeasy Mini Kit (QIAGEN). The quality of the total RNA was checked using an Agilent 2200 TapeStation (Agilent Technologies, Santa Clara, CA). Then, 50 ng of total RNA of each gene was used for the labeling reaction with the one-color protocol of an Agilent Low-Input QuickAmp Labeling Kit (Agilent Technologies). Labeled RNA was hybridized to a 60K Agilent 60-mer SurePrint technology (SurePrint G3 Mouse Gene Expression 8 × 60K Microarray Kit version 2.0) according to the manufacturer's protocol. All hybridized microarray slides were washed and scanned using an Agilent scanner. Relative hybridization intensities and background hybridization values were calculated using Agilent Feature Extraction software (version 9.5.1.1). Raw signal intensities and flags for each probe were calculated from hybridization intensities and spot information according to procedures recommended by Agilent. The raw signal intensities of two samples were $\log_2$-transformed and normalized using a quantile algorithm in the 'preprocessCore' library package of the Bioconductor software (http://www.bioconductor.org/). This produced a gene expression matrix consisting of 55,681 probe sets; differentially expressed genes between samples were selected using a Z-score of 2.0 or more and a ratio of 1.5-fold or more. For downregulated genes, a Z-score of −2.0 or less and a ratio of 0.75 or less were used. The full data have been deposited in National Center for Biotechnology Information gene expression omnibus (Accession#:GSE103598).

## miRNA microarray analysis

ALDH-positive and -negative 4T1 cells were separated by ALDEFLUOR assay as described above. Total RNA was extracted from cells using RNAiso. The quality of the total RNA was checked using an Agilent 2200 TapeStation (Agilent Technologies, Palo Alto, USA). Then, 1000 ng of total RNA was hybridized, washed, and scanned by the Flash Tag Biotin HSR RNA Labeling Kit according to the manufacturer's protocol. In total, 3163 miRNAs were scanned on the GeneChip miRNA 4.0 Array (Affymetrix, Santa Clara, CA) and data were analyzed by Affymetrix Transcriptome Analysis Console 3.0. Differentially expressed miRNAs between samples were selected by calculating ratios as follows: for upregulated miRNAs, a ratio of twofold or more; for downregulated miRNAs, a ratio of 0.5-fold or less. The full data have been deposited in National Center for Biotechnology Information gene expression omnibus (Accession#:GSE157655).

## Treatment of miR-182::hairpin inhibitor

Hairpin inhibitor of miR-182 (Horizon Discovery Ltd., Cambridge, UK) was transfected using Lipofectamine 2000 reagent (Invitrogen) according to the manufacturer's instructions. 48 hr after transfection, cells were harvested, and subjected to Dual-Luciferase reporter assay (Promega).

## Preparation of mmu-miR-182 knockout 4T1 cells

Genomic editing and knockout of miR-182 were performed using the CRISPR/Cas9 system. sgRNA targeting a proximal region of mmu-miR-182–5 p was constructed by the Guide-it sgRNA In Vitro Transcription Kit (Takara Bio). The primer sequence was as follows: 5'-CCTCTAATACGACTCACTA TAGGCAATGGTAGAACTCACACGTTTAAGAGCTATGC-3'. Guide-it Recombinant Cas9 (Electroporation-ready) (Takara Bio) and sgRNA were co-transfected by electroporation, and single cells were seeded by FACS Aria. Each clone was cultured, and mutations in genomic DNA were detected by Sanger sequencing. Prediction of sgRNA off-target effects was performed using Off-Spotter (*Pliatsika and Rigoutsos, 2015*). Three bases of mismatches with sgRNA are detected in eight alleles, and any mutations were not confirmed in these alleles.

## Statistical and data analyses

JMP software was used to perform statistical analyses. The values presented are expressed as the mean with SD. All experiments were performed at least in triplicate. The significance of differences between two groups was analyzed by two-tailed Student's t-tests while those with greater than two groups were analysis of variance (ANOVA), followed by Tukey–Kramer post hoc tests or Dunnett's test. Equal variances were not formally tested. A 5% level of probability was considered significant. No statistical method was used to predetermine sample sizes; however, our sample sizes were similar to those reported in previous studies (*Koyanagi et al., 2016*; *Matsunaga et al., 2018*).

## Acknowledgements

This research was supported by Platform Project for Supporting Drug Discovery and Life Science Research (Basis for Supporting Innovative Drug Discovery and Life Science Research (BINDS)) from the Japan Agency for Medical Research and Development (AMED). VECELL 3D plates were gifted by Makoto Kodama, PhD (VECELL, Inc). We are grateful for the technical support provided by the Research Support Center, Graduate School of Medical Sciences, Kyushu University.

## Additional information

### Funding

| Funder | Grant reference number | Author |
|---|---|---|
| Ministry of Education, Culture, Sports, Science and Technology | Grant-in-Aid for Scientific Research A, 16H02636 | Shigehiro Ohdo |
| Ministry of Education, Culture, Sports, Science and Technology | Challenging Exploratory Research, 17H06262 | Shigehiro Ohdo |
| Ministry of Education, Culture, Sports, Science and Technology | Challenging Exploratory Research, 20K21484 | Satoru Koyanagi |
| Ministry of Education, Culture, Sports, Science and Technology | Challenging Exploratory Research, 20K21901 | Naoya Matsunaga |
| Ministry of Education, Culture, Sports, Science and Technology | Scientific Research B, 18H03192 | Naoya Matsunaga |
| Ministry of Education, Culture, Sports, Science and Technology | Specially Promoted Research, 17H06096 | Yoshitaka Fukada |

| Ministry of Education, Culture, Sports, Science and Technology | Scientific Research B, 25440041 | Hikari Yoshitane |
| Japan Agency for Medical Research and Development | PRIME, 17937210 | Hikari Yoshitane |
| Ministry of Education, Culture, Sports, Science and Technology | JSPSKAKENHI Grant, 17J01969 | Takashi Ogino |
| Japan Agency for Medical Research and Development | JP20am0101091 | Shigehiro Ohdo |
| Japan Agency for Medical Research and Development | JP21am0101091 | Shigehiro Ohdo |

The funders had no role in study design, data collection and interpretation, or the decision to submit the work for publication.

## Author contributions

Takashi Ogino, Resources, Data curation, Formal analysis, Funding acquisition, Validation, Investigation, Visualization, Methodology, Writing - original draft; Naoya Matsunaga, Conceptualization, Resources, Data curation, Formal analysis, Supervision, Funding acquisition, Validation, Investigation, Methodology, Project administration, Writing - review and editing; Takahiro Tanaka, Resources, Data curation, Formal analysis, Validation, Visualization, Methodology; Tomohito Tanihara, Investigation; Hideki Terajima, Resources, Methodology; Hikari Yoshitane, Yoshitaka Fukada, Resources, Funding acquisition, Methodology, Writing - review and editing; Akito Tsuruta, Resources, Supervision, Validation; Satoru Koyanagi, Conceptualization, Resources, Supervision, Funding acquisition, Validation, Project administration, Writing - review and editing; Shigehiro Ohdo, Conceptualization, Supervision, Funding acquisition, Project administration, Writing - review and editing

## Author ORCIDs

Naoya Matsunaga (iD) https://orcid.org/0000-0002-0004-4535
Hikari Yoshitane (iD) http://orcid.org/0000-0001-6319-3354
Shigehiro Ohdo (iD) https://orcid.org/0000-0003-4795-9764

## Ethics

Animal experimentation: All experimental procedures were performed after approval and following the guidelines of Kyushu University (approval number: A20-131-0).

## Decision letter and Author response

Decision letter https://doi.org/10.7554/eLife.66155.sa1
Author response https://doi.org/10.7554/eLife.66155.sa2

# Additional files

## Supplementary files

• Supplementary file 1. Primer sequences for quantitative PCR analysis.

• Supplementary file 2. The mRNA levels of transcription factors estimated to bind to genomic. *Clock* promoter regions in microarray analysis of ALDH-positive and -negative 4T1 cells.

• Supplementary file 3. The expression of miRNAs reported as regulating. *Clock* expression in ALDH-positive and -negative 4T1 cells in microarray analysis.

• Transparent reporting form

## Data availability

The full data of microarray analysis have been deposited in National Center for Biotechnology Information gene expression omnibus (miRNA microarray, accession#:GSE157655; mRNA microarray,

accession#:GSE103598). All data generated or analysed during this study are included in the manuscript and supporting files. Source data files of the quantitative data have been provided for all figures.

The following previously published dataset was used:

| Author(s) | Year | Dataset title | Dataset URL | Database and Identifier |
|---|---|---|---|---|
| Matsunaga N, Ogino T, Hara Y, Tanaka T, Koyanagi S, Ohdo S | 2018 | ALDH high or low cell in 4T1 cell | https://www.ncbi.nlm.nih.gov/geo/query/acc.cgi?acc=GSE103598 | NCBI Gene Expression Omnibus, GSE103598 |

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
