## [Decision Letter]

**Acceptance summary:**

Cancer stem cells (CSCs) function as the root of recurrence and metastasis in breast cancer. Therefore, targeting CSCs may improve breast cancer prognosis. The findings of the work focus on the regulation circadian component CLOCK on CSC and present a possible strategy to overcome the malignancy of BCSCs by targeting miRNA-mediated post-transcriptional regulation, which propose an innovative impact on the novel rationales to eradicate CSCs.

**Decision letter after peer review:**

Thank you for submitting your article "Post-transcriptional repression of circadian component CLOCK regulates cancer-stemness in murine breast cancer cells" for consideration by *eLife*. Your article has been reviewed by 3 peer reviewers, including Caigang Liu as the Reviewing Editor and Reviewer #1, and the evaluation has been overseen by Mone Zaidi as the Senior Editor.

Summary:

The authors studied the relationship between CLOCK gene and ALDH expression in BCSCs and provided a potential strategy targeting miR-182 for management of breast cancer. The findings are interesting and the major conclusions are justified by the current work.

Essential Revisions:

1. In figure 1, the impact of high ALDH expression on the circadian rhythm needs to be included though has been previously reported. Correlation between CLOCK and ALDH and clinical prognosis should be discussed and presented in Supplemental Figures to provide more convincible results. Detailed data on metastasis to different organs in 4T1 PDX model would better to be supplemented in figure 4 and figure 7.

2. How many alleles were shown for the miR-182 CRISPR editing and how was this obtained? Were the sequences of the indels in all alleles established and what effort was made to rule out off-target events? All of the above should be described in Materials and methods.

3. Statistical methods were not mentioned, neither were number of replicates, animals and animal sex described in the Methods and Figure Legends. The description of Results should be written more and ALDH should be discussed deeper on its potential direct regulation relationship with CLOCK. The limits of the current study would better to be mentioned in Discussion.

*Reviewer #1:*

In this article the authors elucidated that one of the circadian components, CLOCK, functions as the regulator of ALDH, whose high expression characterizes the stemness of breast cancer cells. Also, the post-transcription regulation by miR-182 was also revealed to further discover the upstream regulator of CLOCK and proposed that repression of miR-182 may be a potent avenue for breast cancer therapy. The authors employed 4T1 cell line and murine PDX model to study the impact of CLOCK on the expression of ALDH by measurement of several markers reflecting the stemness and proliferative and metastatic ability of breast cancer cells, thus explained one possible mechanism which associate the disruption of circadian rhythm with the stemness of BCSC. The topic was relative innovative as the field of relationship between circadian rhythm and tumor biology has attracted more intense interest recently. The whole study was logically designed and the amount of work completed by the authors is basically abundant to justify the conclusion. However, if further evaluation on the detailed mechanism of interaction between CLOCK and ALDH be supplemented would be more rigorous. Additionally, only miR-182 was evaluated for its clinical significance, the clinical data about the relationship between expression of CLOCK and ALDH and the outcome of patients is warranted.

Strengths:

1. On the cellular experiment level, the authors firstly transfected 4T1 cell with five candidate circadian gene, among which overexpression of CLOCK has the most prominent impact on decline of ALDH-positive cells. Thus, further studies were launched to examine the effect of CLOCK on ALDH expression level and transcription activity of CLOCK in ALDH-positive cells, explaining the negative regulation between CLOCK and ALDH. Also, CLOCK-transduced 4T1 cells were evaluated for its transcriptional level of 3 stemness-related genes and growth rate and spheroid formation ability was compared with mock-transduced cells, the results all indicate that high expression of CLOCK correlates with enhanced stemness property and proliferation rate of tumor cell. in vitro measurement of tumor cell migration was conducted to mimic and dynamically monitor the metastatic process by 3D cell culture chip and spheroid invasion assay, which are all fresh new measurement method that objectively reflect the decreased invasive potential of CLOCK-transduced cells, the metastatic capacity was quantified by testing the expression level of EMT process related genes. All of the above work is abundant to support the negative regulation of CLOCK on ALDH, along with the biological property of tumor cells.

2. Additional in vivo experiments were conducted with 4T1 bearing PDX murine model and the CLOCK-expressing group display increased growth rate and more pulmonary metastatic foci.

3. By comparing the difference in Luciferase mRNA level, the authors proved that the regulation of CLOCK expression is in a post-transcription manner by miRNA binding to 3'UTR of CLOCK gene instead of during the transcription process. Subsequently miRNA microarray analysis was conducted and miR-182 was assessed as the miRNA regulating CLOCK expression. Level of miR-182 is higher in ALDH-positive cells and knockout of which in ALDH-positive cells result in higher clock mRNA level, in vivo experiments also shown decreased malignancy in miR-182 knockout 4T1 cells. All of the above findings indicate the negative role of miR-182 on CLOCK expression and repression of miR-182 show promise for breast cancer treatment.

Weaknesses:

1. The CLOCK expression level displays a negative correlation with ALDH, while the precise mechanism of CLOCK regulating ALDH or the interaction between CLOCK and ALDH are not mentioned.

2. Insufficient in vive experiment data regarding metastasis in different organs in figure 4 and figure 7, metastatic foci on other organs should be included to enrich the in vivo experiment data.

3. Clinical relevance of miR-182 was evaluated with data source from public database, however the impact of expression level of CLOCK and ALDH on outcome of patients was not mentioned, this part of data and analysis is important to prove the clinical significance.

Comments for authors:

1. The CLOCK expression level displays a negative correlation with ALDH, while the precise mechanism of CLOCK regulating ALDH or the interaction between CLOCK and ALDH are not mentioned. As is referred to in the paper CLOCK functions as transcription factor, figure out the regulation process of CLOCK on the transcription process of downstream structural genes, thus impact on ALDH expression would be more rigorous.

2. In figure 4 and figure 7, the in vivo study of 4T1 PDX model only contains pulmonary metastasis, bone and liver metastasis also constitute the major role in breast cancer metastasis, so metastatic foci on other organs should be included to enrich the in vivo experiment data.

3. Clinical relevance of miR-182 was evaluated with data source from public database, however the impact of expression level of CLOCK and ALDH on outcome of patients was not mentioned, this part of data and analysis is important to prove the clinical significance.

*Reviewer #2:*

This manuscript investigates functional role of aldehyde dehydrogenase (ALDH) and Clock protein in breast cancer stem-like cells (BCSCs). The authors find a possible target: CLOCK, which should be repressed in high-ALDH-activity breast cancer 4T1 cells and abrogated tumor growth and invasive potential. A new microRNA, miR-182, was found for post-transcriptional regulation of CLOCK.

This study is of interest but it lacks depth so, if the authors are able to provide a more concrete and complete analysis, the paper should be accepted.

1. How many alleles were shown for the miR-182 CRISPR editing (Fig7), how was this obtained? Were the sequences of the indels in all alleles established? What effort was made to rule out off-target events? (e.g. sequencing of potential off target candidates).

2. In Page 6 lines 174, the promoter activity of CLOCK::Luc was detected. However, in the Figure 5A the mRNA level of luciferase was detected. It was confused for readers which method was used in Figure 5A? What internal control (b-Actin or *Renilla*) was measured for the mRNA level of luciferase in Figure 5A and 5B, there was no shown in Method and Figure Legends.

3. The statistical description is preliminary. What statistics used was unknown in the Methods. How many replicates were not mentioned in the Figure Legend? Number of animals, animal sex were not mentioned in the Methods and Figure Legend.

4. Although it is empirically believed that the relevant processing does not affect the expression of internal reference genes such as GAPDH, in order to remind the rigor of the data, the authors should analyze the transcriptome of GAPDH under the conditions in the manuscript.

5. The results description and Discussion should be written more. ALDH should be more discussed.

6. Clock gene levels should be tested in the miR-182 KO xenograft and lung metastasis cancer samples.

*Reviewer #3:*

This manuscript is of potential interest to readers in the field of cancer stem cells and circadian. The data quality is high. And the findings are interesting and supporting data are solid.

Deregulation of the circadian in cancer cells is implicated in tumor malignancy. In this manuscript, Ogino et al. showed that repression of CLOCK regulates cancer-stemness in murine breast cancer cells. Their previous study demonstrated that the cells with high ALDH exhibit circadian oscillation. In this study, they demonstrated that the expression of CLOCK was repressed in high-ALDH cancer cells, and enhancement of CLOCK expression abrogated the stemness properties of cancer cells. Furthermore, miR-182 was identified to control the degradation of CLOCK mRNA. Knockout of miR-182 restored the expression of CLOCK, leading to decreased tumor growth.

[Editors' note: further revisions were suggested prior to acceptance, as described below.]

Thank you for submitting your article "Post-transcriptional repression of circadian component CLOCK regulates cancer-stemness in murine breast cancer cells" for consideration by *eLife*. Your article has been reviewed by 2 peer reviewers, including Caigang Liu as the Reviewing Editor and Reviewer #1, and the evaluation has been overseen by Mone Zaidi as the Senior Editor. The following individual involved in review of your submission has agreed to reveal their identity: Qingkai Yang (Reviewer #3).

Essential revisions:

1. It will be better to add the limits of the study in the discussion.

2. Specify how to selectively target the CLOCK 3'UTR of clock (line 355).

3. Please format the manuscript according to the requirements of *eLife* journal.

Summary:

The authors supplemented the experimental data, more detailed description of method to firmly hold back the conclusion that Clock could repress transcription of ALDH, and targeting miR-182 which function as the upstream negative regulator of Clock could be a potential therapeutic target. The most prominent strength was to propose one possible mechanism which mediates Clock mediating downstream ALDH transcription, though concluded from computational prediction and deserves further validation. The results support their conclusion, which will pose an innovative impact on the novel rationales to eradicate cancer stem cells.

*Reviewer #1:*

1. Since the pulmonary tropism of 4T1 cells and low rate of metastasis to other organs have been explained, you may consider to add this to Discussion as one more limit of the study and employ multiple cell lines in your future research.

2. Formal of the revised manuscript seems have not fully meet the requirements of *eLife* journal regarding revised submission, please make corresponding adjustments.

*Reviewer #3:*

In this revised manuscript, the authors have conducted a set of new experiments and added additional data, and have addressed reviewers' comments and also revised the manuscript accordingly. As a result, the quality of this paper has been further improved.

---

## [Author Response]

Reviewer #1:[…] 1. The CLOCK expression level displays a negative correlation with ALDH, while the precise mechanism of CLOCK regulating ALDH or the interaction between CLOCK and ALDH are not mentioned. As is referred to in the paper CLOCK functions as transcription factor, figure out the regulation process of CLOCK on the transcription process of downstream structural genes, thus impact on ALDH expression would be more rigorous.

As reviewer’s valuable comment, this point is very important concern. Thus, we conducted further investigation about the suppressing mechanisms of ALDH activity induced by enhanced-CLOCK expression. CLOCK is a major transcriptional activator, so we assumed that there are some transcriptional repressors mediating *Aldh3a1* suppression. Analysis of the promoter region of *Aldh3a1* gene identified CCAAT/enhancer binding protein α (C/EBPα) as a mediator of CLOCK-induced ALDH suppression. Probably, the attenuation of malignant properties of 4T1 was not fully explained only by C/EBPα; however, this mediator should have important roles for this attenuation. These data were presented as Figure 2—figure supplement 1 of revised manuscript. Descriptions about these data were incorporated into Result and Discussion section as follows:

Result section (Page5, line132 – 140.)

“Database analysis of the mouse *Aldh3a1* gene upstream region led to identification of CCAAT/enhancer binding protein α (C/EBPα) as a mediator of CLOCK-controlled expression of ALDH (Figure 2—figure supplement 1A). […] These results suggest that C/EBPα has important roles on CLOCK-mediated ALDH suppression, through transcriptional repression of *Aldh3a1* gene in 4T1 cells.”

Discussion section (Page8, line260 – 267.)

“C/EBPα was identified as a mediator of CLOCK-induced ALDH suppression. Reports suggest that up-regulation of C/EBPα contributes breast tumors suppression through induction of apoptosis and inhibition of cell proliferation (Lourenço and Coffer, 2017). […]. However, further investigation is required to evaluate whether this CLOCK-induced ALDH suppression can generalized to all types of cancer cells because the function of C/EBPα is altered by organs, differentiation state of cells, and transcriptional co-regulators (Lourenço and Coffer, 2017; Ossipow et al., 1993; Ramji and Foka, 2002).”

2. Insufficient in vive experiment data regarding metastasis in different organs in figure 4 and figure 7, metastatic foci on other organs should be included to enrich the in vivo experiment data.

4T1 breast cancer cells metastasize to mainly lung, but also liver, brain, and bone (Heppner et al., 2000). Therefore, we evaluated the number of metastatic foci on these organs. Only few tumor colonies were detected in lung, and not any tumor colonies were detected in brain in this experiment. Tumor colonies were detected from bone, so we incorporated these data into Figure 4D and Figure 7C of revised manuscript. Descriptions about these data were incorporated into Result and Materials and methods section as follows:

Result section (Page6, line170 and Page7, line232 – 233.)

“Moreover, CLOCK-expressing 4T1 cell-bearing mice had limited formation of tumor colonies in lung and bone marrow (*P* <0.01, Figure 4C and 4D). Limited number of tumor colonies were observed in miR-182 KO cells implanted-mice lung (*P*<0.01, Figure 7B); also in bone marrow (*P*<0.01, Figure 7C).”

Materials and methods section (Page18, line483 – 488.)

“To isolate metastatic tumor colony in bone marrow, bone marrow cells were collected from mouse femora and treated with RBC lysis buffer (155 mM NH4Cl, 10 mM NaHCO3, 0.1 mM EDTA). […] Tumor colonies were fixed with methanol, then stained by 0.2% crystalbiolet for quantification.”

3. Clinical relevance of miR-182 was evaluated with data source from public database, however the impact of expression level of CLOCK and ALDH on outcome of patients was not mentioned, this part of data and analysis is important to prove the clinical significance.

As suggested, clinical relevance of CLOCK or ALDH is important for our manuscript. ALDHs are referred to as a marker of poor prognosis in several reports(Charafe-Jauffret et al., 2010; Marcato et al., 2011). Also, disruption of Clock genes is reported as tumor promoter (Cadenas et al., 2014). However, there are both positive and negative clinical relevance for Clock expression in clinical database. This both clinical relevance may be due to these clinical data were obtained from heterogenic tumor cells. Therefore, we revised descriptions as follows with Figure 7—figure supplement 4:

Discussion section (Page10, line336 – 339.)

“The amino acid sequence and function of CLOCK is highly conserved among species; thus, similar effects can be expected in human malignant breast cancers. […] To assess the true relevance to patient prognosis, evaluation of CLOCK expression levels in CSCs may be needed.”

Reviewer #2:This manuscript investigates functional role of aldehyde dehydrogenase (ALDH) and Clock protein in breast cancer stem-like cells (BCSCs). The authors find a possible target: CLOCK, which should be repressed in high-ALDH-activity breast cancer 4T1 cells and abrogated tumor growth and invasive potential. A new microRNA, miR-182, was found for post-transcriptional regulation of CLOCK.This study is of interest but it lacks depth so, if the authors are able to provide a more concrete and complete analysis, the paper should be accepted.1. How many alleles were shown for the miR-182 CRISPR editing (Fig7), how was this obtained? Were the sequences of the indels in all alleles established? What effort was made to rule out off-target events? (e.g. sequencing of potential off target candidates).

Three bases of mismatches with sgRNA were detected in Chromosome 1, 5, 8, 9, 10, 14, 15, and 16 using Off-Spotter software. We performed Sanger sequencing analysis, and any mutations were not confirmed in these alleles. Therefore, we revised descriptions as follows:

Materials and methods section (Page20 – 21, line559 – 562.)

“Prediction of sgRNA off-target effects was performed using Off-Spotter (Pliatsika and Rigoutsos, 2015). Three bases of mismatches with sgRNA are detected in eight alleles, and any mutations were not confirmed in these alleles.”

2. In Page 6 lines 174, the promoter activity of CLOCK::Luc was detected. However, in the Figure 5A the mRNA level of luciferase was detected. It was confused for readers which method was used in Figure 5A? What internal control (b-Actin or Renilla) was measured for the mRNA level of luciferase in Figure 5A and 5B, there was no shown in Method and Figure Legends.

As reviewer’s valuable comments, our manuscript was lacking Methods about transcriptional- or post transcriptional regulation of *Clock* mRNA. In the experiment about promoter activity using *CLOCK*::Luc (Figure 5A), *Neomycin phosphotransferase* (*Neo^r^*) derived from *CLOCK*::Luc vector was assessed to normalize transfection efficiency. On the other hand, in the post transcriptional assay using each CLOCK 3’UTR reporter (Figure 5B), the expression levels of *RLuc* mRNA derived from co-transfected pRL-SV40 vector were assessed for normalization. These information were incorporated into Materials and methods section, and Result sections was revised as follows:

Result section (Page6, line180.)

“The expression levels of *Luciferase* mRNA derived from *Clock*::Luc in ALDH-positive 4T1 cells was slightly, but significantly, higher than in ALDH-negative cells (*P*<0.01, Figure 5A right).”

Materials and methods section (Page19, line502 – 511.)

“Transcriptional and post-transcriptional assay of *Clock* mRNA.

*Clock*::Luc or each *Clock* 3’UTR luciferase construct were transfected using lipofectamine LTX reagent according to the manufacturer’s instructions. […] In post-transcriptional assay, the expression levels of *RLuc* mRNA derived from co-transfected pRL-SV40 vector (Promega) for normalization.”

3. The statistical description is preliminary. What statistics used was unknown in the Methods. How many replicates were not mentioned in the Figure Legend? Number of animals, animal sex were not mentioned in the Methods and Figure Legend.

All experiments were performed at least in triplicate. Two-tailed Student’s t tests were applied in analysis of two groups, and analysis of variance (ANOVA), followed by Tukey–Kramer post-hoc tests or Dunnett’s test were applied in analysis of more than three groups. These statistical analyses were performed using JMP software. These information were incorporated into Materials and methods section.

Materials and methods section (Page21, line565 – 572.)

“JMP software was used to perform statistical analyses. […] The significance of differences between two groups was analyzed by two-tailed Student’s t tests while those with greater than two groups were analysis of variance (ANOVA), followed by Tukey–Kramer post-hoc tests or Dunnett’s test.”

For animal experiments, female Balb/c mice were used in all experiments, and number of animals were described in each figure legend.

4. Although it is empirically believed that the relevant processing does not affect the expression of internal reference genes such as GAPDH, in order to remind the rigor of the data, the authors should analyze the transcriptome of GAPDH under the conditions in the manuscript.

As suggested, some reports indicated that the expression of internal reference genes such as GAPDH showed circadian oscillation (Shinohara et al., 1998). Therefore, we preliminary confirmed the expression of three internal reference genes, *18s rRNA*, *β-Actin, and Gapdh under the conditions: enhanced-CLOCK expression or miR-182 knockout.* The expressions of *18s rRNA* and *β-Actin were stably detected; however, the expression levels of Gapdh were difficult to detect stably in 4T1 cells. Moreover, mRNA microarray data (*Accession#: GSE103598, in this manuscript*) show no alteration of expression levels in 18s rRNA and β-Actin between ALDH-positive and -negative cells, so we applied 18s rRNA or β-Actin to normalize qPCR data.*

5. The results description and Discussion should be written more. ALDH should be more discussed.

*According to reviewer’s valuable comment, we conducted further* investigation about the interaction between CLOCK and ALDH. Database analysis of promoter region of Aldh3a1 identified CCAAT/enhancer binding protein α (C/EBPα) as a mediator of CLOCK-induced ALDH suppression. C/EBPα is reported as breast tumor suppressor, and highly expression is detected well-differentiated cells. Consequently, High ALDH activity in cancer stem cells may be due to low expression of CLOCK and C/EBPα, resulted in poor prognosis of patient. These data were presented as Figure 2—figure supplement 1 of revised manuscript. Descriptions about these data were incorporated into Result and Discussion section as follows:

Result section (Page5, line132 – 140.)

“Database analysis of the mouse *Aldh3a1* gene upstream region led to identification of CCAAT/enhancer binding protein α (C/EBPα) as a mediator of CLOCK-controlled expression of ALDH (Figure 2—figure supplement 1A). […] These results suggest that C/EBPα has important roles on CLOCK-mediated ALDH suppression, through transcriptional repression of *Aldh3a1* gene in 4T1 cells.”

Discussion section (Page8, line257 – 267.)

“High activity of ALDH correlates with tumor grade, metastasis, and poor prognosis in patient breast tumor (Charafe-Jauffret et al., 2010; Marcato et al., 2011); consequently, low expression levels of CLOCK in ALDH-positive 4T1 cells likely promote their malignant properties by regulating ALDH activity. […] ALDH suppression can generalized to all types of cancer cells because the function of C/EBPα is altered by organs, differentiation state of cells, and transcriptional co-regulators (Lourenço and Coffer, 2017; Ossipow et al., 1993; Ramji and Foka, 2002).”

6. Clock gene levels should be tested in the miR-182 KO xenograft and lung metastasis cancer samples.

As suggested, we confirmed the expression levels of *Clock* gene in the miR-182 KO xenograft both in whole tumor cells and ALDH-sorted tumor cells. The *Clock* levels were increased by miR-182 knockout, and the effect was drastically in ALDH-positive cells. In lung metastasis cancer samples, it was difficult to evaluate because the number of tumor colonies were fewer in miR-182 KO 4T1 implanted mice lung than naive 4T1 at the same point. These data were presented as Figure 7—figure supplement 1 of revised manuscript. Descriptions about these data were incorporated into Result section as follows:

Result section (Page7, line228 – 231.)

“The expression levels of Clock mRNA in resected tumor were significantly increased by knockout of miR-182 (Figure 7—figure supplement 1A); and the effect was more potent in ALDH-positive cell (Figure 7—figure supplement 1B).”

[Editors' note: further revisions were suggested prior to acceptance, as described below.]

Reviewer #1:1. Since the pulmonary tropism of 4T1 cells and low rate of metastasis to other organs have been explained, you may consider to add this to Discussion as one more limit of the study and employ multiple cell lines in your future research.

According to your valuable comment, we added about pulmonary tropism of 4T1 in Discussion section, and the metastatic capacity to other organs is unclear point. As your suggestion, we should investigate these points using multiple cell lines in our future research. Thus, we revised manuscript as follows in Discussion section:

Discussion section (Page10, line305 – 310.)

“4T1 is lung tropic cancer cells (Monteran et al., 2020); thus, metastatic colonies were detected from mainly lung in this study. […] To evaluate the effect of CLOCK expression on metastatic capacities to multiple organs, further investigations are also needed to employ several cancer cells.”

2. Formal of the revised manuscript seems have not fully meet the requirements of eLife journal regarding revised submission, please make corresponding adjustments.

As suggested, we revised our manuscript according to *eLife* editorial support. We uploaded Figures, Figure supplements, and Supplementary files individually, and listed at the end of article files.